# An integrated spatio-temporal view of riverine biodiversity using environmental DNA metabarcoding

William Bernard Perry [1,2,18] ✉, Mathew Seymour [3,18] ✉, Luisa Orsini [4], Ifan Bryn Jâms[2], Nigel Milner[1], François Edwards [5], Rachel Harvey[6], Mark de Bruyn [7], Iliana Bista[8,9,10,11], Kerry Walsh[12], Bridget Emmett[6], Rosetta Blackman[13,14,15], Florian Altermatt [13,14], Lori Lawson Handley[15], Elvira Mächler[13], Kristy Deiner [16], Holly M. Bik[17], Gary Carvalho [1], John Colbourne[4], Bernard Jack Cosby[6], Isabelle Durance [2] & Simon Creer [1] ✉

Anthropogenically forced changes in global freshwater biodiversity demand more efficient monitoring approaches. Consequently, environmental DNA (eDNA) analysis is enabling ecosystem-scale biodiversity assessment, yet the appropriate spatio-temporal resolution of robust biodiversity assessment remains ambiguous. Here, using intensive, spatio-temporal eDNA sampling across space (five rivers in Europe and North America, with an upper range of 20–35 km between samples), time (19 timepoints between 2017 and 2018) and environmental conditions (river flow, pH, conductivity, temperature and rainfall), we characterise the resolution at which information on diversity across the animal kingdom can be gathered from rivers using eDNA. In space, beta diversity was mainly dictated by turnover, on a scale of tens of kilometres, highlighting that diversity measures are not confounded by eDNA from upstream. Fish communities showed nested assemblages along some rivers, coinciding with habitat use. Across time, seasonal life history events, including salmon and eel migration, were detected. Finally, effects of environmental conditions were taxon-specific, reflecting habitat filtering of communities rather than effects on DNA molecules. We conclude that riverine eDNA metabarcoding can measure biodiversity at spatio-temporal scales relevant to species and community ecology, demonstrating its utility in delivering insights into river community ecology during a time of environmental change.

Freshwater biodiversity has been in sharp decline during the 20th and 21st centuries due to multiple anthropogenic pressures, with monitored freshwater populations showing an 83% average decline between 1970 and 2018[1]. The scale and pace of decline requires immediate action, with biodiversity monitoring being key in informing policy[2]. Although information on abiotic properties of ecosystems are accessible at fine spatio-temporal scales, information on biodiversity is not[3]. Comprehensive knowledge of freshwater aquatic biodiversity underpins the effectiveness of habitat management, restoration, and conservation[4].

Environmental DNA (eDNA) analysis, which utilises DNA (intracellular and extracellular) of unicellular and multicellular organisms, and their gametes, has proven a powerful biodiversity monitoring method in freshwaters[5]. Advantages include non-destructive sampling, scalable technologies, automation and wide taxonomic coverage at

potentially unprecedented spatio-temporal scales; qualities that compliment many non-molecular approaches, while also providing greater sensitivity in several cases[6,7]. It has therefore been highlighted as fundamental to addressing the needs of future biodiversity conservation[8] and ecology[3].

There are, however, properties of eDNA which require greater understanding, including fundamental questions of its origin, state, fate and transport, the so called 'ecology of eDNA'[9]. Specific to rivers, the persistence of eDNA is determined by the balance between the fate of eDNA (i.e., decay rate) and the distance of downstream transport[10]. These factors dictate whether eDNA-inferred measures of biodiversity are dominated by transport, resulting in nested biodiversity in samples from the lower catchment[11]. Conversely, rapid decay would result in little downstream transport, and a turnover of biodiversity, allowing accurate species detection on localised scales. Transport and degradation of eDNA have been assessed in several independent studies with varying results, with detection 240 m[12] to 100 km[13] downstream of the source. Yet, how transport and degradation impacts biodiversity measures at different spatial and temporal scales in rivers remains ambiguous, limiting our understanding of whether ecological communities detected are present at a site. Understanding the spatio-temporal resolution of biodiversity measures, which eDNA analysis can provide, requires multi-taxa, high-resolution, spatio-temporal sampling in well-characterised ecosystems.

Here, we intensively sampled 14 sites along the River Conwy, a well-documented lake-fed river in Wales (UK)[14], at 19 timepoints over a year (April 2017 to April 2018), each with three 1 L sample replicates. Sampling of three additional lake-fed rivers in Europe (Tywi, Gwash and Glatt) and one in North America (Skaneateles Creek) was carried out at one timepoint. Diversity was assessed across the metazoan tree of life using three genetic markers, each offering identification of different taxa. In the Conwy, exogenous mackerel DNA was also released at the source of the river to measure eDNA transport downstream. Patterns of alpha and beta diversity along the rivers reflected ecological communities across space, time and environmental conditions without being confounded by variation introduced by eDNA, such as through eDNA transport. eDNA, therefore, has the potential to provide an ecosystem-wide view of diversity, with a broad array of novel ecological insights from taxa, which are often overlooked in non-molecular biodiversity assessments.

## Results

A total of 798 water samples were taken from 14 sites and 19 timepoints (27 April 2017 to 18 April 2018) along the River Conwy (Wales, UK). In addition to these samples, in July 2017, 39 samples were collected along the River Glatt (Switzerland), 33 samples were collected along the River Gwash (England, UK), 36 samples were collected along the River Tywi (Wales, UK) and 33 samples were collected along the Skaneateles Creek (USA). A total of 896 samples were successfully sequenced, producing 178,833,278 12S (average of 199,591 per sample), 144,016,790 18S (average of 160,733 per sample) and 279,175,484 COI (average of 311,580 per sample) single reads.

### Spatial variation

The Atlantic mackerel (*Scomber scombrus*), used as an introduced source of eDNA at the outlet of Llyn Conwy (the source of the River Conwy), was detected at sites E01, E02, E03, E04 and E05 (Fig. 1a) a total of 30 times across the study with the 12 S marker, reaching a maximum transport of 5000 m downstream of the release site. Although, in most cases, it was not detected beyond 1000 m downstream (Supplementary Fig. 1). Mackerel was detected further downstream at sites closer to the estuary (E11, E12 and E14), but these occurrences have been treated as originating from natural sources. Using the COI marker, the mackerel was detected at sites E01, E02 and E03, reaching a maximum transport of 250 m downstream of the release site.

The River Conwy was split into upper (sites E01–E05), middle (sites E06–E10), and lower sections (sites E11–E14) based on important environmental characteristics (Supplementary Fig. 2) such as water pH and conductivity, as well as land use type. eDNA data indicated a 37.5% overlap of fish species detected in the upper, middle, and lower sections of the river (Figs. 2a), and 87.5% of all fish species were detected in the lower section. There was less overlap in metazoan amplicon sequencing variants (ASVs) between the three river sections (Fig. 2b), with 10.0% of ASVs detected in all three sections. Aquatic arthropods

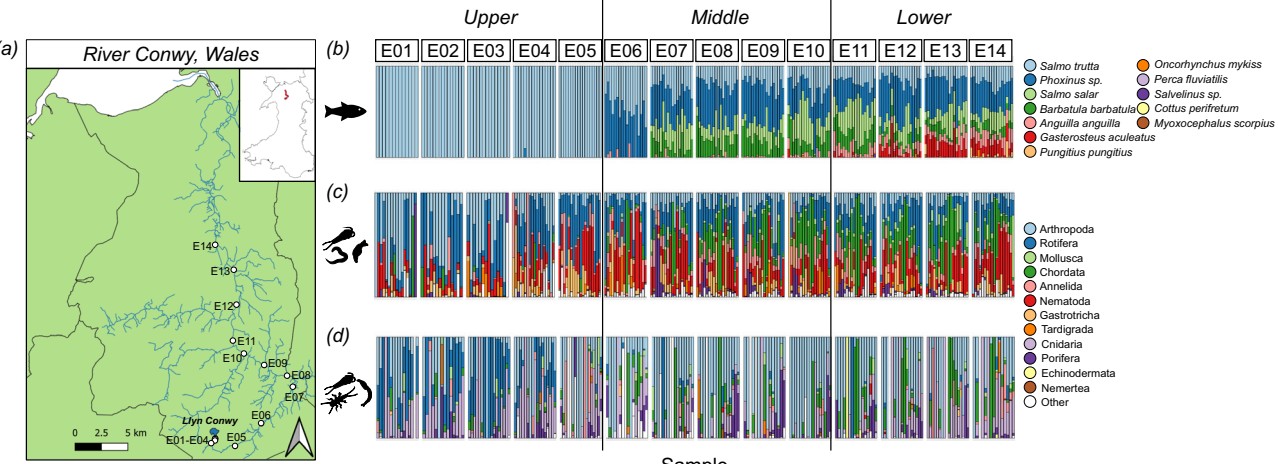

**Fig. 1 | Sample sites across the River Conwy (Wales, UK) with corresponding taxonomic overview.** A simplified catchment map of the **a** River Conwy, Wales, showing sampling sites as white dots (E01–E14) that were sampled at 19 timepoints for eDNA over the course of a year. The same sample sites are shown in the inset map of Wales as red dots. Corresponding stacked bar plots show normalised reads at 19 time points, grouped by sample sites, and coloured by taxonomy, featuring **b** fish detected with the 12 S marker, **c** metazoans detected with the 18S marker and **d** aquatic arthropods detected with the COI marker. Taxonomic identification is shown at the species level for the 12S marker, and at the phylum level for the 18S

and COI markers. White bars represent lower read count phylum and the colour order of the key matches the colour order of the stacked bars. Sites in the stacked bar plots are separated into the upper, middle and lower sections of the river. Source data are provided as a Source Data file. The maps in (**a**) contain OS data © Crown copyright and database right 2023 as well as Natural Resources Wales information © Natural Resources Wales and Database Right. All Rights Reserved. Contains Ordnance Survey Data. Ordnance Survey Licence number AC0000849444. Crown Copyright and Database Right.

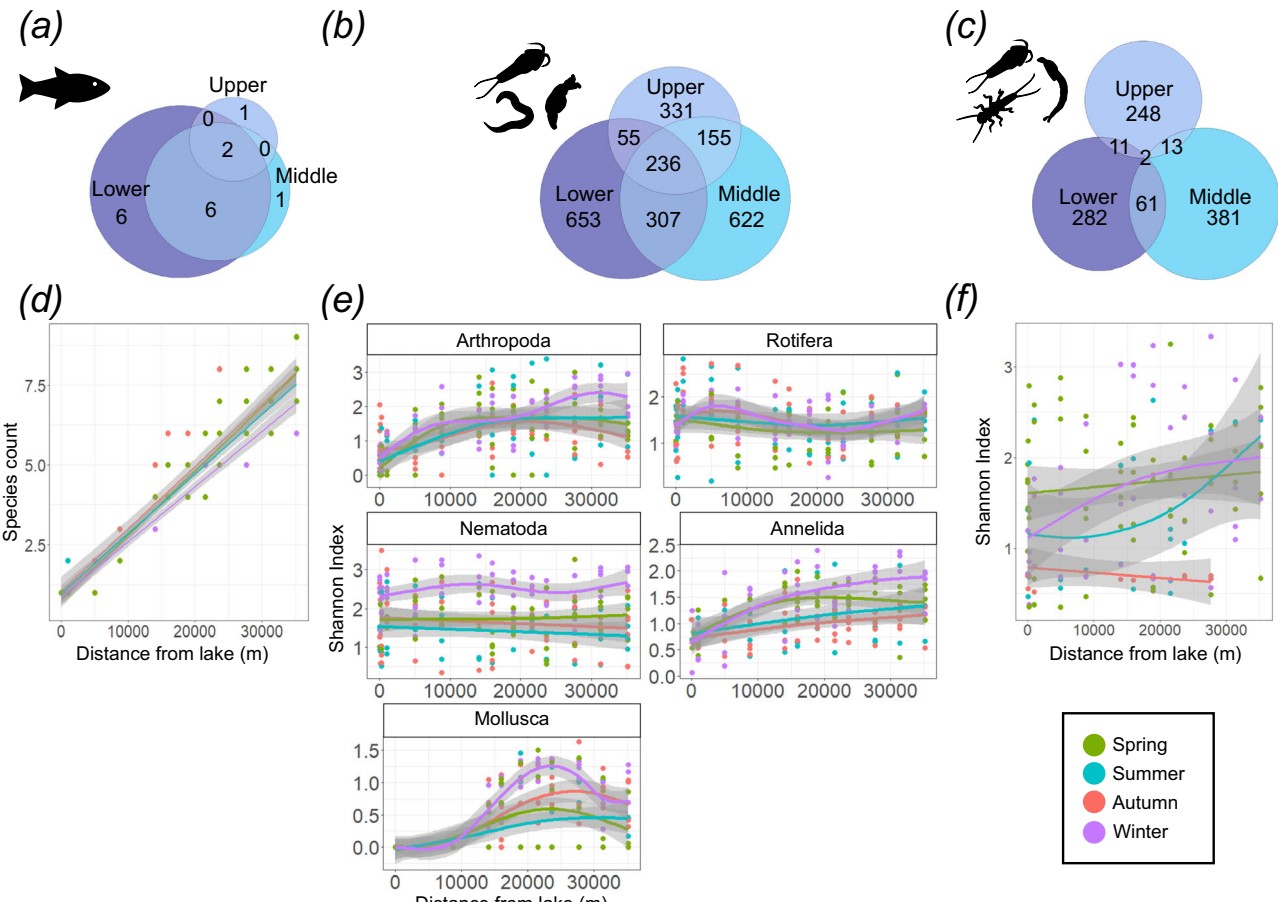

**Fig. 2 | Spatiotemporal variation in measures of alpha diversity across the River Conwy (Wales, UK).** Venn diagrams showing the overlap in **a** fish species (12S marker), **b** metazoan ASVs (18S marker) and **c** aquatic arthropod ASVs (COI marker) between the upper (E01–E05), middle (E06–E10) and lower (E11–E14) sections of the River Conwy. Segments of the river are based on changing environmental characteristics and land use at different points in the river (Supplementary Fig. 3). The size of the circles are scaled to the number of species/ASVs that they represent.

Also shown are alpha diversity plots of **d** fish species count, **e** Shannon index of the most abundant metazoan phyla (annelids, arthropods, nematodes, molluscs and rotifers) and **f** Shannon index of aquatic arthropods. Alpha diversity is shown over distance from the lake, coloured by season, with smoothed conditional means and 95% grey confidence intervals provided by linear models (**d**, **f**) and (**e**) generalised additive models. Source data are provided as a Source Data file.

showed the least amount of overlap, with only 0.2% of ASVs detected in all three sections (Fig. 2c).

In the Conwy, distance from the source lake (Llyn Conwy) had a significant impact on fish species count (Fig. 2d) and Shannon index of aquatic arthropod (Fig. 2f), total arthropod, annelid and mollusc (Fig. 2e) ASVs, but not for rotifers or nematodes (Supplementary Table 2). In the cross-river comparison of the five rivers, river sampled had a significant impact on fish species count ($F_{3,34} = 91.70$, $p < 0.01$) (Fig. 3a) and Shannon index of metazoans ($F_{4,49} = 9.41$, $p < 0.01$) (Fig. 3b) and aquatic arthropod ASVs ($F_{4,32} = 14.32$, $p < 0.01$) (Fig. 3c). Across the rivers, distance from the lake had a significant impact on fish species count ($F_{1,34} = 7.56$, $p < 0.01$) and Shannon index of metazoans ($F_{1,49} = 27.16$, $p < 0.01$) and aquatic arthropod ASVs ($F_{1,32} = 21.96$, $p < 0.01$). Additionally, there was a significant interaction term between distance from the lake and river sampled for fish species count ($F_{3,34} = 4.51$, $p < 0.01$) and Shannon index of metazoans ($F_{4,49} = 2.50$, $p = 0.05$) and aquatic arthropod ASVs ($F_{4,32} = 2.77$, $p = 0.04$).

In the Conwy, distance from the lake had a significant impact on beta diversity of fish species (Fig. 4a, b), aquatic arthropods (Fig. 4e, f), total arthropod, annelid and rotifer ASVs, but not nematodes or molluscs (Fig. 4d). In the cross-river comparison, river sampled had a significant impact on beta diversity of fish species ($F_{4,50} = 31.51$, $p < 0.01$) (Fig. 3e), metazoans ($F_{4,57} = 5.82$, $p < 0.01$) (Fig. 3f) and aquatic arthropods ($F_{4,46} = 1.29$, $p < 0.01$) (Fig. 3g). Across the rivers, distance from the

lake had a significant impact on the beta diversity of fish species ($F_{1,50} = 21.96$, $p < 0.01$), metazoans ($F_{1,57} = 4.04$, $p < 0.01$) and aquatic arthropods ($F_{1,46} = 1.13$, $p = 0.02$). Additionally, there was a significant interaction term between distance from the lake and river sampled for beta diversity of fish species ($F_{4,50} = 6.69$, $p < 0.01$), metazoans ($F_{4,57} = 2.22$, $p < 0.01$) and aquatic arthropods ($F_{4,46} = 1.11$, $p < 0.01$).

In the Conwy, samples collected spatially close together had lower dissimilarity than samples collected further apart, with dissimilarity increasing with distance between samples, and this was seen across each of the taxonomic groups (Fig. 5a–c), indicating that communities change along the course of the river. Increasing spatial dissimilarity in fish communities was due to nestedness (Fig. 5d), whereas, for metazoans and aquatic arthropods, it was due to turnover (Fig. 5g). In the cross-river comparison, nestedness was also seen to be important in the fish species of the River Tywi (Fig. 6d), however, turnover was more important in the Rivers Gwash and Glatt (Fig. 6g). For metazoans and aquatic arthropods, a consistent trend was seen across all rivers, with high turnover (Fig. 6h) and low nestedness (Fig. 6e).

## Temporal variation
In the Conwy, in line with expectations, the season had a significant impact on fish species count (Fig. 2d) as well as the Shannon index for aquatic arthropod (Fig. 2f), nematode, annelid and mollusc (Fig. 2e) ASVs, but not the Shannon index of total arthropod or rotifer ASVs

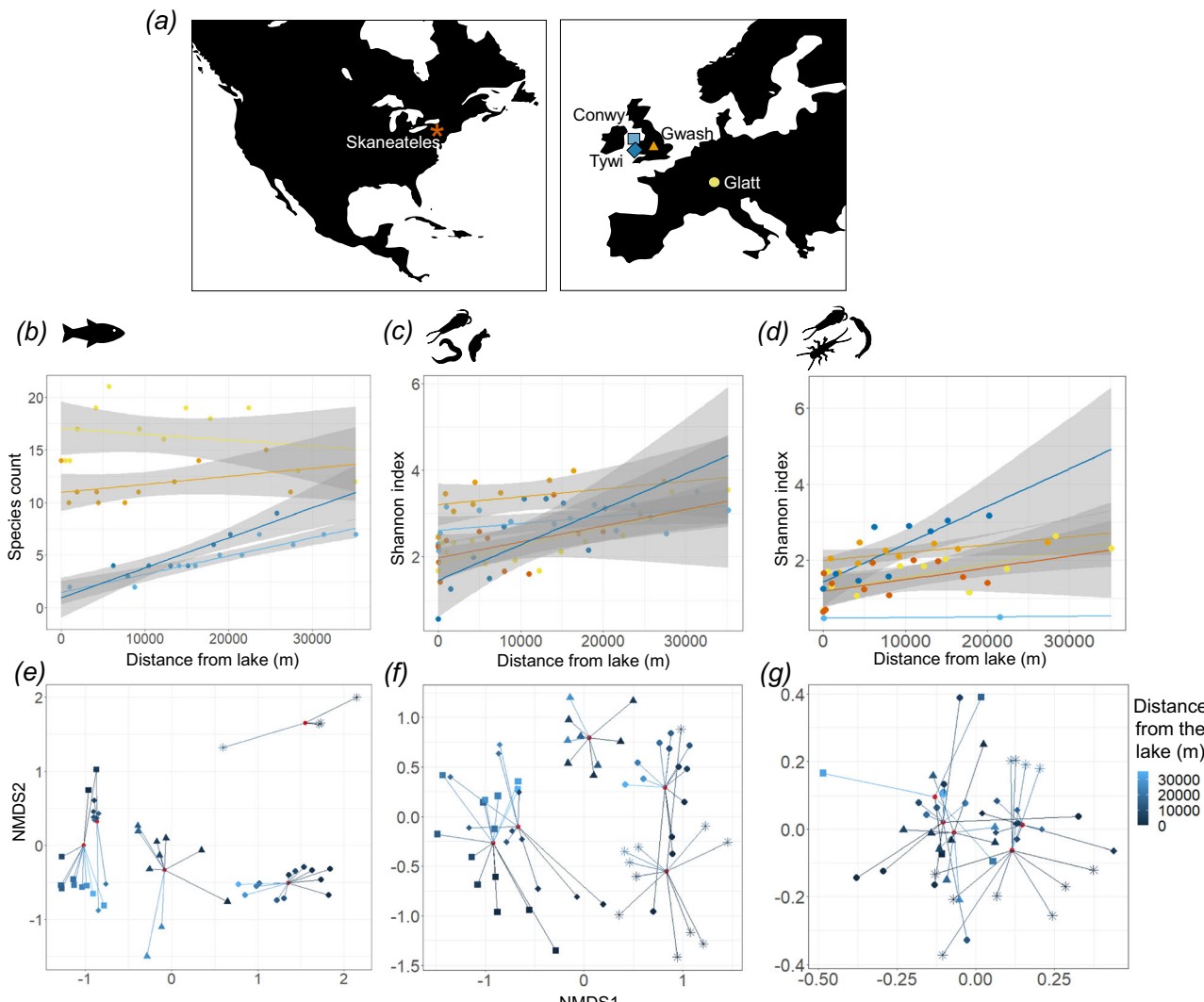

**Fig. 3 | Alpha and beta diversity within each of the five rivers, the Rivers Conwy (Wales, UK), Tywi (Wales, UK), Gwash (England, UK), Glatt (Switzerland) and Skaneateles Creek (USA). a** Maps of North America and Europe show the geographic location of the rivers and highlight the colour and shape coordination used in (**b**–**d**). Alpha diversity is represented by (**b**) species count of fish detected with the 12S marker (Skaneateles Creek was excluded as only a few samples passed the filtering criterion), **c** Shannon index of metazoan ASVs detected with the 18S marker and (**d**) Shannon index of aquatic arthropod ASVs detected with the COI marker, all of which are plotted against distance from the lake and contain a linear regression with corresponding 95% confidence intervals (grey). NMDS plots of beta diversity using **e** fish species, **f** 18S metazoan ASVs and **g** COI aquatic arthropod ASVs, broken down by distance from the lake (colour) and river sampled (shape), with red dots representing the mean NMDS scores for each of the rivers. Source data are provided as a Source Data file.

(Supplementary Table 2). However, there was a significant interaction term between season and distance from the lake for fish species count and Shannon index of aquatic arthropod, total arthropod, nematode and annelid ASVs (Supplementary Table 2). The rotifers were the only phylum where there was no significant impact of season independently, or through an interaction (Supplementary Table 2).

Season also had an impact on the beta diversity of fish species as well as aquatic arthropod, total arthropod, annelid, nematode and rotifer ASVs, but not mollusc ASVs (Fig. 4d). There was little evidence of an impact of temporal variation on nestedness or turnover for fish species (Fig. 5d, g) or aquatic arthropod (Fig. 5c, f) ASVs in the Conwy, however, temporal variation was detected for metazoan ASVs, which showed reduced turnover in samples that were taken closer together in time (Fig. 5h).

### Environmental conditions

18S metazoan and COI aquatic arthropod ASVs were not clustered by phylogeny in their response to the environmental conditions assessed

(Supplementary Fig. 3). Environmental conditions also had differing impacts on taxonomic groups, with no single variable causing a significant change in alpha diversity across all taxonomic groups (Supplementary Table 2). pH and monthly temperature significantly impacted the alpha diversity of the greatest number of taxonomic groups, although fish and total arthropods were not impacted by pH. Fish and the rotifers were not impacted by monthly temperature variation, although the rotifers were close to being significant. Similarly, for beta diversity, pH and temperature also had the largest impact, with aquatic arthropods being the only taxonomic group not significantly impacted by these two variables.

### Discussion

For eDNA analysis to be an effective biodiversity monitoring tool, ecological patterns must be identifiable despite the effects of eDNA persistence from species that no longer occupy a site. Here, we provide hitherto unreported empirical evidence, at temporally relevant and replicated, catchment-wide scales, that it is possible to detect alpha

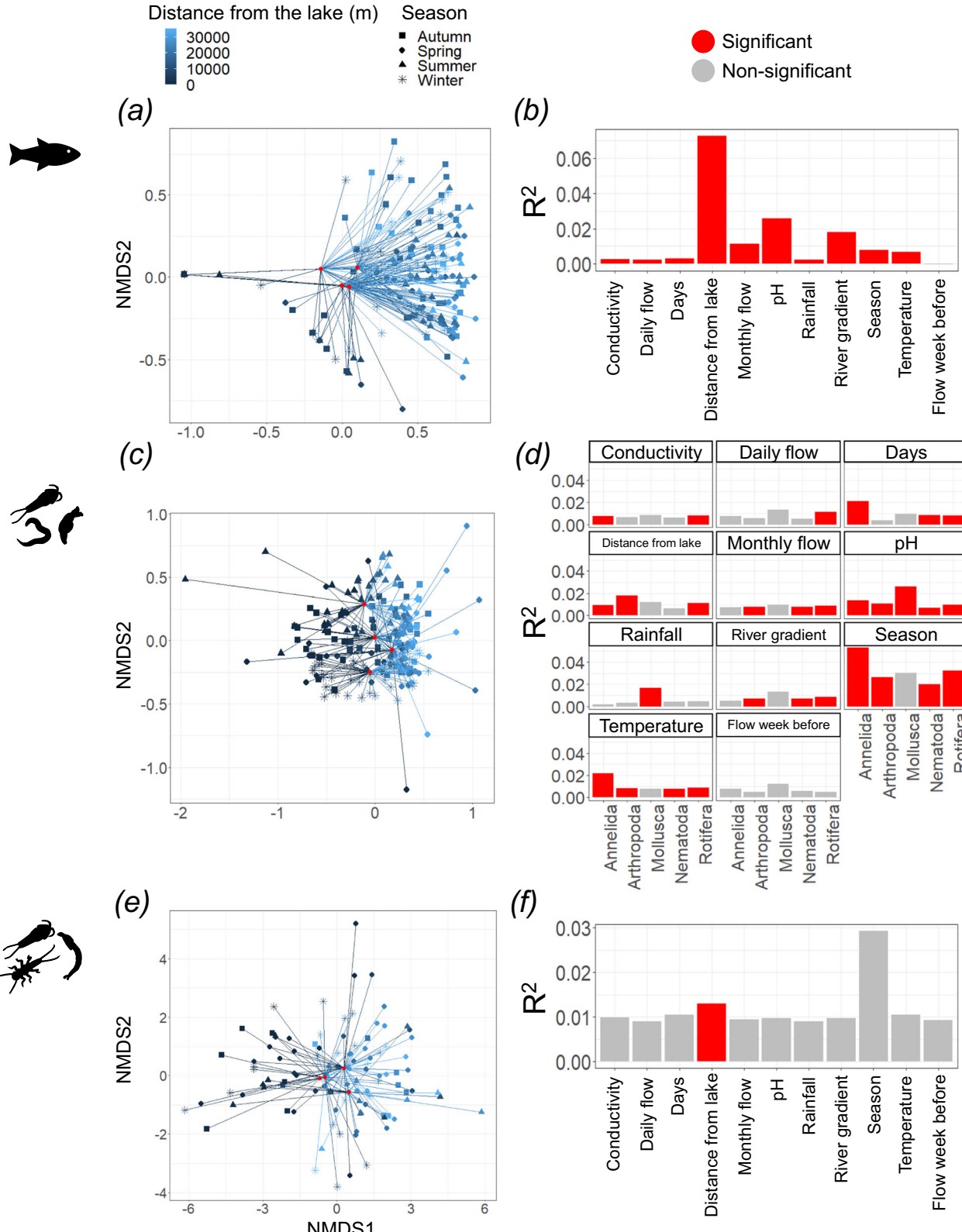

**Fig. 4 | Spatiotemporal variation in measures of beta diversity across the River Conwy (Wales, UK), in addition to the relationship between beta diversity and environmental conditions.** Beta diversity of the River Conwy for **a**, **b** fish species detected with the 12S marker, **c**, **d** metazoan ASVs detected with the 18S marker and **e**, **f** aquatic arthropod ASVs detected by the COI marker. Included are (**a**, **c**, **e**) NMDS plots coloured by distance the sample was taken from the lake (Llyn Conwy), with shapes denoting season, red dots displaying seasonal mean NMDS scores and lines connecting datapoints with their respective seasonal means. $R^2$ values from permutational multivariate analysis of variance (PERMANOVA) are also included for **b** fish, **d** metazoans (split by the five most abundant phyla) and **f** aquatic arthropods. Source data are provided as a Source Data file.

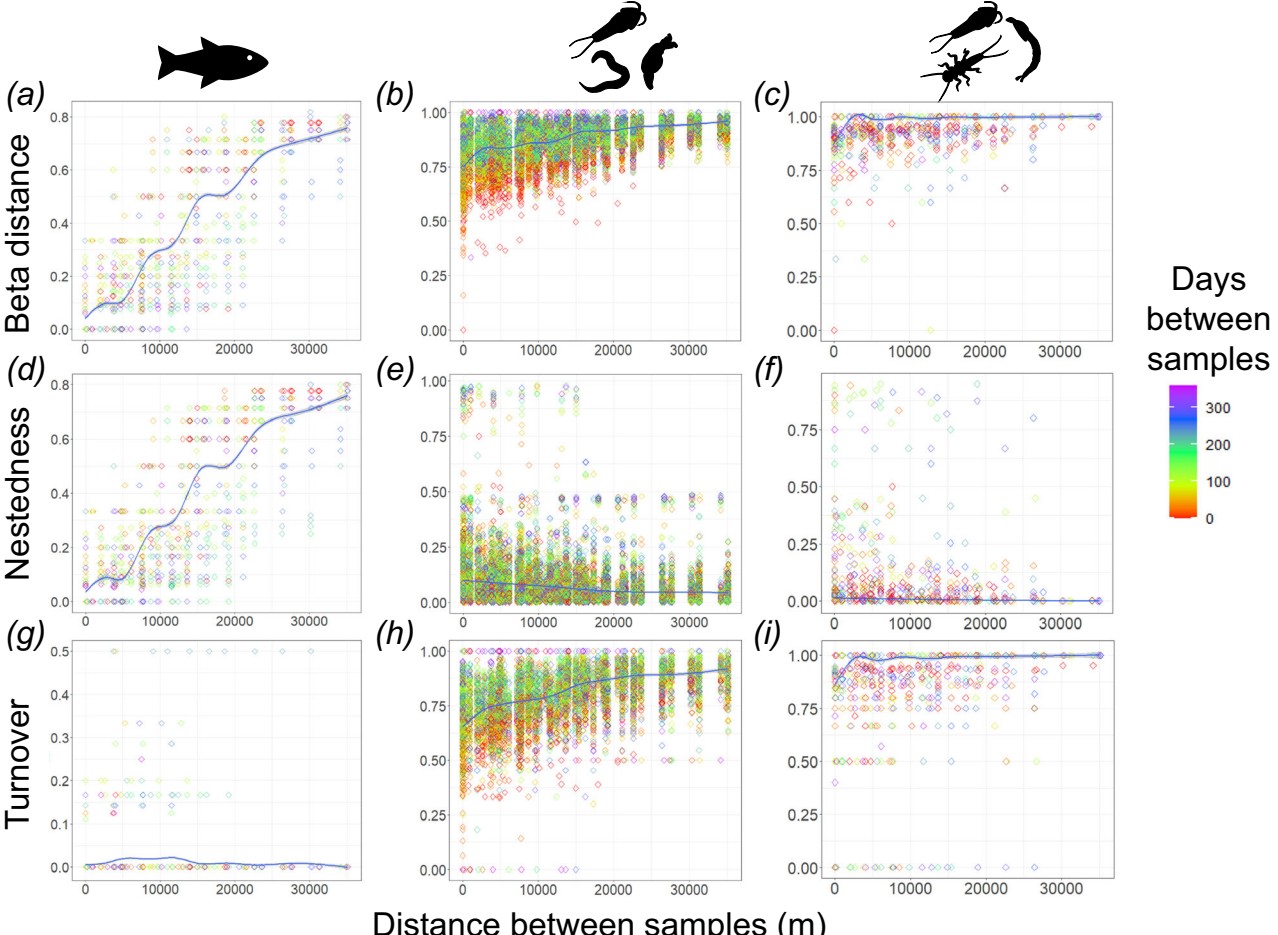

**Fig. 5 | Spatiotemporal variation of pairwise beta diversity in the River Conwy (Wales, UK).** Pairwise beta diversity in the River Conwy, based on (**a**–**c**) Sørensen dissimilarity, including two components, **d**–**f** nestedness and **g**–**i** turnover, are plotted against the pairwise geographic distance between samples for fish species detected with the 12S marker (**a**, **d**, **g**), metazoan ASVs detected with the 18S marker (**b**, **e**, **h**) and aquatic arthropod ASVs detected with the COI marker (**c**, **f**, **i**). Loess smoothers with corresponding 95% confidence intervals (grey) are also present. Datapoints are coloured by pairwise difference in days between sample collections. Source data are provided as a Source Data file.

and beta diversity shifts, along with coherent ecological signals, across a breadth of riverine phyla[13,15–17], thereby informing future study designs and eDNA monitoring programs.

Introduced mackerel eDNA at the Conwy source was detected up to 5000 m downstream, demonstrating relatively localised transport. The upper reaches of the Conwy are characterised by low pH (e.g. site E04 had a pH range of 4.25–5.83) caused by acid moorland, which likely increases eDNA degradation rates, shortening its transport potential. A lotic mesocosm experiment (also in Wales, UK) showed that, in acidic conditions (pH 5.35–5.9), over 90% of eDNA copies were lost within an hour, compared to 3 h in neutral conditions (pH 6.73-6.82)[18]. The Conwy is ~55 km long and takes from 2 to 5 days to drain. Therefore, using decay rates estimated from the mesocosm experiment, eDNA in the Conwy could theoretically be transported ~460–1140 m under acidic conditions. Under neutral conditions, a transport potential of 1380–3420 m would be expected. The 250–5000 m mackerel eDNA transport distance observed in this study is in a similar range to previous experimental results[18], even when using approximate hydrological parameters for the Conwy and a different detection method (metabarcoding, not qPCR). The greater distance of mackerel detection using the 12S marker (172 bp) rather than the COI marker (313 bp) is also concordant with the understanding that longer DNA fragments degrade more quickly than shorter fragments[19–21], making detection less likely. Although, specificity of the 12S marker for

fish could also contribute, as the detection of fish reads represents a much lower percentage of total COI reads.

Significant variation in natural biodiversity over space was evident across rivers, molecular markers, and taxa. The increase in fish alpha diversity downstream (Fig. 2d) reflects known distribution patterns of freshwater fish communities[22–25] and their propensity to be nested[26,27], due to increased availability, size and heterogeneity of habitats[28] coupled with increased accessibility to diadromous and potadromous species. In the cross-river comparison, the Tywi and Conwy showed very similar longitudinal increases in fish species (Fig. 3a); however, the Gwash and Glatt did not. The differences in slopes between rivers may be due to the nature of the feeder lakes and the transport of lentic fish eDNA. The lakes that feed into the Conwy and Tywi rivers have low levels of fish diversity and are dominated by brown trout (*Salmo trutta*), unlike the lakes that feed into the Gwash and Glatt rivers, which have relatively high fish diversity[29,30]. The lentic transport of eDNA from the diverse fish communities is potentially causing the inflated species count at the source of the Gwash and Glatt, highlighting the importance of identifying natural sources of extraneous eDNA into rivers.

There was congruence between 18S and COI markers, with total arthropods and aquatic arthropods showing a significant change in alpha diversity over the course of the river. Some studies using non-molecular methods have shown stability in alpha diversity between

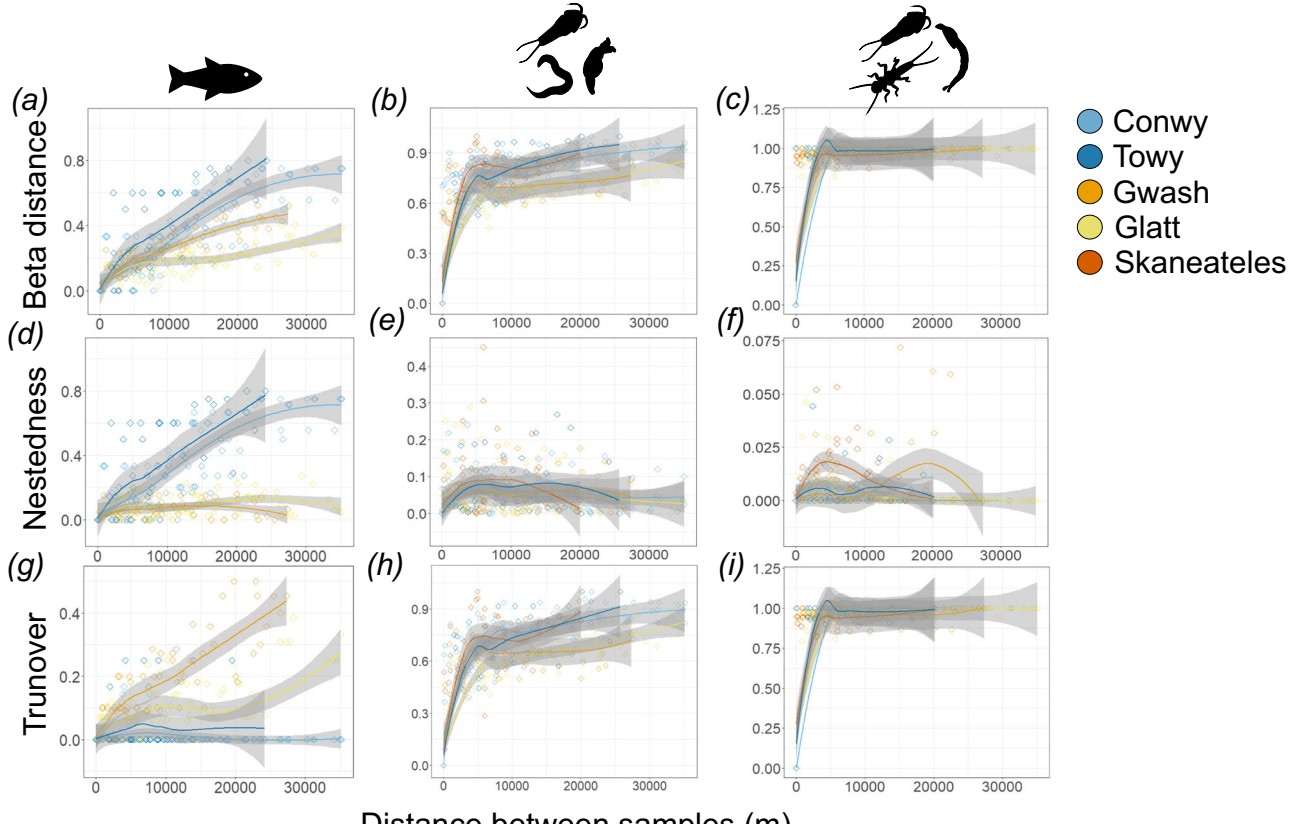

**Fig. 6 | Pairwise beta diversity in five rivers, the Rivers Conwy (Wales, UK), Tywi (Wales, UK), Gwash (England, UK), Glatt (Switzerland) and Skaneateles Creek (USA).** Beta diversity is based on (**a**–**c**) Sørensen dissimilarity, including two components, **d**–**f** nestedness and **g**–**i** turnover, are plotted against the pairwise geographic distance between samples for fish species detected with the 12S marker (**a**, **d**, **g**), metazoan ASVs detected with the 18S marker (**b**, **e**, **h**) and aquatic arthropod ASVs detected with the COI marker (**c**, **f**, **i**). Loess smoothers with corresponding 95% confidence intervals (grey) are also present. Datapoints are coloured according to sample origin. For fish species, Skaneateles Creek was excluded as only a few samples passed the filtering criterion. Source data are provided as a Source Data file.

different sections of a river[31], or decreasing downstream diversity[31], which suggests that eDNA is more effective at capturing a broader range of diversity and thus temporal patterns. Distance from the lake did not have a significant effect on rotifer or nematode alpha diversity, indicating potential species sorting, and reliance on microhabitats which have a consistent carrying capacity of species richness and evenness regardless of position along the river[32,33]. Significant interactions were observed between season and distance from the lake in spring and winter, with two peaks in nematode alpha diversity at 10,000 m and 35,000 m in winter (Fig. 2e), possibly due to high flow-induced transport of nematode eDNA from other parts of the watershed. In the cross-river comparison, distance from the lake also had a significant impact on the alpha diversity of metazoans and aquatic arthropods, with diversity increasing downstream to different extents in each of the rivers (Fig. 3a).

Predictably, beta diversity demonstrated that rivers from separate geographies had more distinct fish and metazoan communities (Fig. 3d–f). However, aquatic arthropod community composition showed overlap between rivers. In the Conwy, there was a significant spatial component to beta diversity seen across most phyla, except for molluscs and nematodes (Fig. 4d). Molluscs were not detected until sample site E07, likely due to the acidic water found in the upper section of the river preventing their establishment. As for nematodes, there was no significant impact of spatial variation on beta diversity, like alpha diversity, and provides further evidence that species sorting and microhabitats available across the entire river may be dictating diversity.

In the cross-river comparison, distance from the lake also had a significant effect on the beta diversity of fish, metazoans, and aquatic arthropods, with its effect size differing between rivers. The largest driver of beta diversity was turnover, observed with fish in the Gwash and Glatt (Fig. 6g), as well as with metazoans (Fig. 6h) and aquatic arthropods (Fig. 6i) in all rivers. Conversely, fish communities in the Conwy and Tywi were nested (Fig. 6d), which could either be due to the transport of eDNA downstream or a reflection of fish community structure. It is unlikely that eDNA transport is the cause, as the same pattern was not seen across all rivers. Instead, it is likely due to the distribution of fish in the river, as nestedness is the predicated structure of fish communities. The role of turnover in fish communities in the River Gwash and River Glatt, and not nestedness, could be due to the short-term transport of lentic eDNA, with lentic species being replaced with resident riverine fish communities. For example, in the Gwash, more lentic species such as European perch (*Perca fluviatilis*), rainbow trout (*Oncorhynchus mykiss*) (stocked in the lake) and Nine-spine stickleback (*Pungitis pungitis*) reduce in relative abundance as you move downstream, whereas more riverine species increase (e.g. European bullhead (*Cottus sp.*))

Temporal change was, overall, less apparent than spatial change. Nonetheless, at least one temporal measure had a significant impact on the alpha diversity of fish, total arthropods, nematodes, annelids, molluscs and aquatic arthropods in the Conwy, with rotifers defying this trend. The greatest fish diversity was detected in autumn and the lowest in winter, driven by the change in detection of species such as Atlantic salmon (*Salmo salar*), three-spined stickleback (*Gasterosteus*

*aculeatus*), non-native rainbow trout (*Oncorhynchus mykiss*), stone loach (*Barbatula barbatula*) and the European eel (*Anguilla anguilla*). The seasonality of some species can be explained by their life cycle (Supplementary Note 1).

Freshwater invertebrates, such as molluscs, annelids and nematodes, had the highest Shannon diversity during winter (Fig. 2e), with increases in freshwater molluscs characterised by a greater detection of the family Sphaeriidae; small freshwater bivalves, indicating that the winter is an important breeding time. The Sphaeriidae are poorly understood, but there is evidence of UK Sphaeriidae releasing broods in November[34]. For annelids, winter was characterised by a greater detection of the family Enchytraeidae (Supplementary Note 2), as well as Lumbriculidae, a freshwater aquatic species. Finally, winter nematode communities were characterised by families Monhysteridae, Teratocephalidae and Tylenchomorpha, which can be explained by their life history (Supplementary Note 3).

Season significantly affected beta diversity for fish, annelids, total arthropods, nematodes and rotifers, but not for molluscs or aquatic arthropods. Temporal changes in fish community composition are driven by the previously discussed migrations, while differences in nestedness and turnover did not show strong temporal changes (Fig. 5d, g). For invertebrates, samples taken closer together in time showed lower differentiation (Fig. 5b) and lower rates of turnover (Fig. 5h), which will contribute to the overall significant seasonal changes in beta diversity, demonstrating the ability for eDNA analyses to detect shifts in community composition over relatively short time periods.

In the Conwy, there was a significant pH effect on the alpha and beta diversity of total arthropods, aquatic arthropods, rotifers, nematodes, annelids and molluscs. pH did not significantly impact the alpha diversity of fish ($F = 3.20$, $p = 0.07$), but did significantly impact the beta diversity of fish. The almost ubiquitous effect of pH on biodiversity is congruent with its known strong effect on aquatic organisms[35–37]. It is also likely that acidity accelerates eDNA degradation[18], and thus impacts diversity metrics. Organisms such as rotifers showed increased diversity in acidic water. That is concordant with the ecology of rotifers, which have been seen to have higher diversity in acidic waters[38]. Evidence of the effect of pH on organism distribution, rather than eDNA degradation, can also be seen in the effect size of pH on the beta diversity of different taxonomic groups. For example, the largest effect size was seen in molluscs (Fig. 4d), with pH being a well-known driver of freshwater mollusc assemblages[39,40] due to their calcium carbonate shells.

Every taxonomic group was significantly impacted by at least one metric of river flow. Fish alpha and beta diversity were impacted by river gradient, a proxy for flow velocity, with river gradient having the third largest effect on beta diversity (Fig. 4d), consistent with our understanding of fish movement and distribution. For example, steep river gradients can represent impassable barriers to migration, such as the Pystyll y Pandy waterfalls between sites E06 and E07 preventing upstream Atlantic salmon and stone loach[41] migration (Fig. 1b). River gradient also had a significant impact on the alpha (Supplementary Table 2) and beta diversity of rotifers and nematodes (Fig. 4d). High flow velocities can destabilise sediments and biofilms, especially during periods of flood, upon which rotifer and nematode communities depend on[42]. All flow measures had a significant impact on nematode alpha diversity, which is concordant with what is already known about the release of nematodes into the water column in times of high flow[43]. Flow during the week before sampling was seen to have a significant impact on the alpha diversity of nematodes and annelids (Supplementary Table 2), an important consideration for eDNA study designs, however, reassuringly, it did not significantly impact beta diversity.

Although temperature has been shown to degrade eDNA[44], in this study, diversity did not show a ubiquitous significant temperature response across phyla. Nematode alpha diversity exhibited a significant response to temperature, with higher temperatures exhibiting lower nematode diversity. Nematodes species have different preferred

temperatures to avoid niche overlap, therefore, temperature changes will induce community change[45]. Aquatic arthropods also displayed a significant response to temperature, showing a normal distribution (peak at ~7.5 °C). Beta diversity of annelids, arthropods, nematodes and rotifers was significantly impacted by temperature, the same phyla whose beta diversity was also significantly impacted by the season, indicating a temperature-driven seasonal effect on community assemblage[21,46].

The sparse partial least squares (sPLS) analyses demonstrated that for fish (Supplementary Fig. 3a), some of the more environmentally responsive species included European eel (*Anguilla anguilla*), Atlantic salmon (*Salmo salar*), stone loach (*Barbatula barbatula*), common minnow (*Phoxinus phoxinus*) and brown trout (*Salmo trutta*). For the metazoans and aquatic arthropods (Supplementary Fig. 3b, c), no single phyla or family, respectively, showed universal sensitivity to environmental variables, highlighting the importance of broad taxonomic coverage in monitoring programmes. Across all taxonomic groups, river gradient, rainfall and temperature showed the least amount of correlation across taxa.

A key highlight from our work is that comprehensive sampling of riverine eDNA can provide ecologically relevant snapshots of diversity on practical spatio-temporal scales without confounding effects of upstream biological communities. The relatively short transport distance of eDNA (i.e. a maximum of 5000 m in the mackerel experiment) means that detectable diversity patterns, especially using 12S and 18S markers, are congruent with established ecological trends yielded via conventional, but more costly, low throughput non-molecular approaches. Cross-river comparisons among rivers from Europe and North America showed that trends in alpha and beta diversity along the river were largely consistent, and in accordance with the ecological characteristics of those rivers. Due to the ability of eDNA to detect changes in organisms across the tree of life, its comprehensive approach can be embraced and interpreted independently from the insights gained from non-molecular approaches[47]. We would, therefore, encourage progressive dialogue between researchers and stakeholders to enhance the standardisation of eDNA metabarcoding approaches for whole ecosystem biodiversity assessment. A multi-taxon approach leverages large-scale, cost-effective biodiversity assessments, providing a holistic view of shifting, taxonomically diverse, riverine communities in freshwater ecosystems.

## Methods
### Sampling
Three replicate water samples (1 L per sample) were collected at each sample site and time point. Sample sites were arranged in a linear longitudinal transect along the River Conwy (14 sample sites, over a 35.2 km stretch of river in Wales (UK), sampled between 27 April 2017 to 18 April 2018) (Fig. 1a), River Tywi (12 sample sites, over a 25.7 km stretch of river in Wales (UK), sampled on 13 July 2017), and River Gwash (11 sample sites, over a 27.4 km stretch of river in England (UK), sampled on 31 July 2017), River Glatt (13 sample sites, over a 35.1 km stretch of river in Switzerland, sampled on 3 July 2017) and Skaneateles Creek, USA (11 sample sites, over a 20 km stretch of river in the USA, sampled on 19 July 2017) (Supplementary Table 1). These rivers were chosen as a representation of lake-fed rivers of a similar size in the northern hemisphere. A total of 939 samples were taken. Samples were collected starting at the source of a river, all of which were lakes, moving downstream into the main river channel until reaching a hydrological endpoint such as an estuary (in the case of the Conwy), or a confluence with another large river. Temporally, all sites in the River Conwy were sampled 19 times through 2017 and 2018 to capture seasonal changes, and the other rivers were sampled once in July 2017. Water samples were filtered through 0.22-µm SterivexTM filter units (EMD Millipore Corporation) using a Geopump TM Series II peristaltic pump (Geotech). In the field, 66 negative controls were taken using deionized water and

were treated the same as the other samples through downstream processing. Also processed and sequenced were 68 laboratory-negative controls. All pre-PCR steps were performed in a PCR-free, eDNA clean room, in a separate building where the PCRs were undertaken. Access to the clean room is restricted to trained users, and the laboratory is regularly cleaned with bleach. Those using the PCR-free room wear PCR-free overcoats, hair nets, shoes, gloves and masks.

### Introduced extraneous eDNA

To introduce a simple and traceable exogenous source of eDNA without risks associated with the introduction of non-native species, five dead fresh Atlantic mackerel (*Scomber scombrus*) (~1.5 kg at any one time) were placed in a plastic mesh bag at the head of the River Conwy, between the exit point of Llyn Conwy and the first sampling point. Three more whole mackerel (~1 kg) were added to the bag every 6 weeks (every other sampling timepoint) to replace degraded mackerel biomass. No material was manually removed from the bag. The total mass during the experiment would have been 2–3 kg.

### DNA extraction and library preparation

DNA was extracted from Sterivex filters using a modified QIAGEN DNA blood and tissue extraction protocol[48], followed by the removal of any potential inhibitors using a QIAGEN Power Clean kit. PCR amplification of the 12S rRNA gene, 18S rRNA gene and mitochondrial cytochrome oxidase subunit 1 (COI) gene were performed using the following primers: 12S MiFish-U forward (5′-GTCGGTAAAACTCGTGCCAGC-3) and reverse (5′-CATAGTGGGGTATCTAATCCCAGTTTG-3)[49], the 18 S TAReuk454FWD1 (5′- 463 CCAGCA(G/C)C(C/T)GCGGTAATTCC-3′) and TAReukREV3 (5′- 464 ACTTTCGTTCTTGAT(C/T)(A/G)A)A-3′)[50] and the COI m1COIintF (5′-GGWACWGGWTGAACWGTWTAYCCYCC-3′) and jgHCO2198 (5′-TAIACYTCIGGRTGICCRAARAAYCA-3′)[51]. A two-step library preparation method was used following Bista et al. (2017), but employing four sets of unique dual indexed 96 tags (*n* = 384) in the second round of PCR to facilitate multiplexing and to reduce cross-talk between samples in downstream analyses as in Brennan et al.[52]. First round PCR was done in triplicate for every sample and each of the three PCR primers, using Q5 HS High-Fidelity mastermix (New England Biolabs) for the 12S and 18S markers, and Thermo Scientific's Ampli-gold mastermix for the COI marker, due to the high number of inosine in the COI primer pair[53]. Triplicates were pooled and underwent second round PCR to add unique dual indexes. The second round PCR used Q5 HS High-Fidelity Master Mix and amplicons from the second round PCR were purified twice using AMPure magnetic beads (Beckman Coulter) and quantitated using a 200 pro plate reader (TECAN) using qubit dsDNA HS solution (Invitrogen). A standard curve was created by running standards of known concentration on each plate against which sample concentration was determined. PCR2 amplicons were mixed in equimolar quantities (at a final concentration of 12 pmol) using a biomek FXp liquid handling robot (Beckman Coulter). The final molarity of the pools was confirmed using a HS D1000 tapestation screentape (Agilent) prior to 250 bp paired-end sequencing on an Illumina HiSeq platform aiming for 100,000 reads per sample and target gene (e.g. 12S, 18S and COI). Library preparation up to round 2 PCR cleaning was performed at Bangor University, assisted by a liquid handling robot (Gilson), prior to round 2 cleaning, pooling and sequencing by EnviSion and BioSequencing at the University of Birmingham (https://www.envision-service.com/).

### Bioinformatics

Once sequences were demultiplexed into samples based on their unique Illumina indexes, further demultiplexing by gene region was achieved using Cutadapt 2.3[54]. Raw FASTQ reads were then fed into the 'DADA2' v 1.14.1 pipeline[55], which allowed for trimming of forward and reverse reads, filtering of erroneous reads according to Illumina error profiles, removal of chimeric reads and merging of forward and

reverse reads. For all gene regions, filtering included a truncation of reads at the first instance of a quality score less than or equal to 2, and a maximum number of expected errors of 2. Forward and reverse reads were trimmed to 150 base pairs for the 12S gene region and 220 base pairs for the 18S and CO1 gene regions.

Of the 134 field and laboratory-negative controls, the majority failed to produce any reads or pass the DADA read quality filtering, read merging and downstream filtering of merged reads. 126 (94%), 92 (69%) and 119 (89%) of the negative controls failed quality control for the 12S, 18S and COI, respectively. Of the remaining reads detected in the negative controls, these were used to filter reads associated with samples using the R package 'microDecon' v 1.0.2[56], except for 12S, where the limited number of taxonomic groups meant that the approach was not suitable.

The average amplicon size produced by the 12S MiFish primers is 172 bp[49], and so to remove larger bacterial sequences, which are also amplified with the 12S primers, amplicon sequence variants (ASVs) that were over 20% longer than this average amplicon size were removed, leaving 2726 ASVs. A broad 'BLAST+' v 2.10.0[57] search was conducted against 12S rRNA gene sequences downloaded from the NCBI nt database[58] in order to identify any non-fish sequences. BLAST was used for taxonomic assignment due to its proven effectiveness compared to more complex approaches[59] as well as its usability. ASVs were removed if they did not meet a percentage identity of 70%, a query cover of 80% and an e value of 1, with similar cut-off parameters having been used previously[60]. Following the broad BLAST, if ASVs were not assigned as bony fish (class Actinopteri), they were removed. The majority of ASVs that were removed were assigned to orders such as frogs (Anura), waterbirds (Pelecaniformes), rodents (Rodentia) and hoofed mammals (Artiodactyla), mainly represented by domestic cows (*Bos taurus*) and sheep (*Ovis aries*). Using the ASVs assigned to the class Actinopteri, a second, more specific BLAST was conducted on the MitoFish database[61] (downloaded 23 September 2020), using a higher threshold of 90% percentage identity, 90% query cover and an e value of 0.001, as has been used previously[62], to ensure accurate taxonomic identification. The first BLAST hit was used for taxonomic assignment. A curation step was also conducted, whereby ASVs that were assigned taxonomy of a species which were not living in the environment, with the eDNA likely occurring due to secondary introduction (e.g. through human consumption, marine species), were removed (Supplementary Table 3). ASVs were then clustered into species based on the assigned taxonomy. If a species had less than 0.05% of the overall sample reads it was removed from that sample, if the ASV had less than an absolute value of 20 it was removed and samples containing less than 1000 reads were also removed.

'SILVAngs' v 1.9.5/1.4.3 (web front-end/analysis pipeline) was used to analyse the 18S sequence data, utilising the SILVA r138.1 database[63]. Using the default settings, the maximum relative amount of ambiguous bases and repeated bases per sequence was set at 2 and 4%, respectively. Using the same threshold as the Silva SSU Parc web database, the minimum alignment identity and alignment score of a sequence to a reference sequence was set at 50 and 40%, respectively. The minimum relative base pair score and minimum relative quality of sequences was set to 30. ASVs were not clustered into operational taxonomic units (OTUs), and so a sequence identity value of 1 was adopted. If an ASV had less than 0.05% of the overall sample reads, it was removed from that sample, and samples containing less than 1000 reads were also removed. Non-metazoans and those metazoans that could not be identified to the phylum level were removed, as well as any sequences that were assigned as primates.

A broad 'BLAST+' v 2.10.0 search was first conducted against COI gene sequences from the Midori database (GenBank release 240)[64], and ASVs were removed if they did not meet an identity of 70% and a query cover of 80%, suggested as a baseline[65]. If an ASV had less than 0.05% of the overall sample reads, it was removed from that sample, and samples

containing less than 1000 reads were also removed. Finally, only metazoan phyla were retained, including Arthropoda, Gastrotricha, Platyhelminthes, Annelida, Chordata, Rotifera, Mollusca, Cnidaria, Nematoda, Tardigrada, Porifera, Placozoa, Onychophora, Nemertea, Echinodermata and Bryozoa. Due to the high taxonomic resolution of the COI marker, arthropods were further classified into aquatic and terrestrial arthropods based on taxonomy, with only aquatic arthropods kept in the analysis to better assess river biodiversity.

Rarefaction was not implemented due to its documented limitations[66], in addition to all samples showing adequate diversity saturation with appropriate read depths achieved (Supplementary Fig. 5). Sample triplicates were aggregated after filtering, giving an average number of reads per sample site at a particular time point. All reads were normalised by dividing the read count per ASV divided by the total read count for that sample.

## Statistical analysis

For the River Conwy, explanatory variables used for modelling included distance the sample was taken from the source lake outlet and river source (referred to as distance from the lake), days after first sample collection, season, an interaction between distance from the lake and season, average monthly air temperature (°C), average daily flow (m³/s), average monthly flow (m³/s), average seasonal flow (m³/s), average flow the week before the sample was taken (m³/s), average monthly rainfall (mm), river gradient (%), conductivity (μS/cm) and pH. Seasons included winter (December–February), spring (March–May), summer (June–August) and autumn (September–November). Average daily flow, average monthly flow, average flow the week before the sample was taken, river gradient and average monthly air temperature were not available for sites E13 and E14. All flow estimates were based on measurements taken from a flow gauge at site E12 at Cwmlanerch (53.106375, −3.7920625) downloaded from the National River Flow Archive, and were estimated for sites upstream, proportional to catchment area. Temperature and rainfall were measured at the Glasgwm Automatic Weather Station (53.029018, −3.8408066). Water conductivity (μs/cm at 25 °C) and pH measurements were taken on the same day as eDNA sampling for all sites. The river gradient was calculated using a 25 m circular buffer around the sampling point. These abiotic factors were chosen due to their importance in driving freshwater community assemblages.

All statistical analyses were carried out in R v. 3.6.2[67] (Supplementary Code 1). 'Vegan' v 2.4.2[68] was used to calculate Shannon index values for the 18S and COI markers using the normalised read counts for the proportion an ASV contributes to a community and the number of ASVs as a proxy for species number. For the River Conwy, linear models and generalised additive models (GAMs) were constructed using the R package 'stats' v 4.1.0[67] and 'mgcv' v 1.8.35[69], respectively. The linear model had the response variable: species count (not based on normalised reads, instead based on presence or absence) from 12S fish ASVs, and the GAMs had the response variables: Shannon index from the 18S metazoan, and COI aquatic arthropod ASVs. Shannon index was used for the latter as the disparity in read counts between ASVs was more marked, along with the samples having much greater diversity. Therefore, information on which ASVs were rare or common in a sample, as provided by the Shannon index, was useful information. All explanatory variables in the GAM were included as smoothing terms, other than season. The variable distance from the lake was also separated by season. Only the variables distance from the source lake, days after the first sample collection and season were included in the 18S mollusc GAM to reduce unique covariate combinations below the specified maximum degrees of freedom. The average flow the week before the sample was taken was not included in the COI GAM to reduce coefficients. The decision to drop this term was based on similar measures already being included in the analysis, such as daily flow. An interaction term between season and distance from the lake

was also included in the 12S linear model. For the linear model and the COI GAM, a stepwise function was used to remove non-significant terms using 'stats' v4.1.0, which implements a backward stepwise search that removed terms to reach the lowest Akaike Information Criterion (AIC). In the cross-river comparison, linear models were utilised for all markers using the same response variables used in the models for the River Conwy. Explanatory variables were a distance from the lake as well as the river sampled, along with an interaction term between these variables. For fish species, Skaneateles Creek was excluded as only a few samples passed the filtering criterion.

'Vegan' v 2.4.2 was also used to assess beta diversity using the 'adonis' function to perform a permutational multivariate analysis of variance (PERMANOVA) under the Bray-Curtis method and 999 permutations, as well as using the metaMDS function to perform Non-metric Multidimensional Scaling (NMDS) under the Bray-Curtis method, 1000 random starts in search of a stable solution and three dimensions. Normalised read count were used as the input for the PERMANOVA and NMDS plots for all markers. Before the beta diversity analysis, sample outliers, as identified by NMDS plots, were removed (Supplementary Table 4). Beta diversity in the form of Sørensen dissimilarity was also calculated and further partitioned into nestedness and species replacement (turnover), derived using the 'betapart' v 1.6 package[70], and was based on presence and absence.

## Reporting summary

Further information on research design is available in the Nature Portfolio Reporting Summary linked to this article.

## Data availability

The metabarcoding data generated in this study have been deposited in the European Nucleotide Archive database under accession code ERP132733 [https://www.ebi.ac.uk/ena/browser/view/PRJEB48362]. The metadata associated with the metabarcoding data used in this study are provided in the Supplementary Data 1. Source data are provided with this paper. The NCBI nt (https://www.ncbi.nlm.nih.gov/nucleotide/), MitoFish (https://mitofish.aori.u-tokyo.ac.jp/download/), SILVA r138.1 (https://www.arb-silva.de/documentation/release-138/) and Midori (https://www.reference-midori.info/) databases were also used in this work. Source data are provided with this paper.

## Code availability

R code used in this study can be found in Supplementary Code 1, with no restrictions.

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

## Acknowledgements

The project was funded by a NERC Directed Highlight Topic Grant (NE/N006216/1) awarded to S.C., with additional grants: NE/N005724/1 awarded to B.J.C., NE/N005716/1 awarded to J.C. and NE/N005678/1 awarded to I.D. We would like to thank the computing facilities provided by Supercomputing Wales, along with the expertise provided by Ade Fewings and Aaron Owen. The high throughput sequencing data were generated by EnviSion, BioSequencing and BioComputing at the University of Birmingham (https://www.envision-service.com/), and we thank Stephen Kissane for technical assistance with liquid handling robotics and high throughput sequencing data generation. Finally, we thank Jose Andrés and Paul Czechowski for assistance in the field to collection samples from Skaneateles Creek.

## Author contributions

K.D., M.d.B., M.S., N.M., F.E., K.W., B.E., H.M.B., J.C., B.J.C., I.D. and S.C. conceived and designed the study. The manuscript was written by W.B.P., with contributions from all co-authors (M.S., L.O., I.B.J., N.M., F.E., R.H., M.d.B., I.B., K.W., B.E., R.B., F.A., L.L.H., E.M., K.D., H.M.B., G.C., J.C., B.J.C., I.D. and S.C.). M.S., R.H., R.B., I.B., B.E, F.A., L.L.H, K.D., G.C. and E.M. were involved in sample collection and the acquisition of data. W.B.P., K.D., M.d.B., L.O., N.M., F.E., I.B.J., H.M.B., J.C., B.J.C., G.C., I.D. and S.C. and were involved in the interpretation of data. M.S. contributed to field sampling design and co-ordinated field and laboratory operations, including DNA extraction and sequencing library preparation in collaboration with I.B. L.O. further contributed to sequencing library preparation and sequencing. Bioinformatics and statistical analyses were carried out by W.B.P., with E.M. contributing to analyses on nestedness and turnover. Meteorological and hydrological metadata on the River Conwy associated with the eDNA samples was provided by B.J.C.

## Competing interests

K.D. is the co-founder of SimplexDNA, a company which sells services for environmental DNA analysis. The remaining authors declare no competing interests.

## Additional information

[1]Molecular Ecology and Evolution at Bangor (MEEB), School of Biological Sciences, Bangor University, Bangor, Gwynedd LL57 2UW, UK. [2]Water Research Institute, Cardiff University, Cardiff CF10 3AX, UK. [3]The University of Hong Kong, Hong Kong SAR, China. [4]Environmental Genomics Group, School of Biosciences, University of Birmingham, Birmingham B15 2TT, UK. [5]APEM Ltd, A17 Embankment Business Park, Heaton Mersey, Manchester SK4 3GN, UK. [6]Centre for Ecology & Hydrology, Environment Centre Wales, Bangor LL57 2UW, UK. [7]Australian Research Centre for Human Evolution, School of Environment and Science, Griffith University, Queensland 4111, Australia. [8]LOEWE Centre for Translational Biodiversity Genomics, 60325 Frankfurt, Germany. [9]Senckenberg Research Institute, 60325 Frankfurt, Germany. [10]Naturalis Biodiversity Center, Darwinweg 2, 2333 Leiden, Netherlands. [11]Wellcome Sanger Institute, Tree of Life, Wellcome Genome Campus, Hinxton CB10 1SA, UK. [12]Environment Agency, Horizon House, Deanery Road, Bristol BS1 5AH, UK. [13]Department of Aquatic Ecology, Eawag: Swiss Federal Institute of Aquatic Science and Technology, Überlandstrasse 133, CH-8600 Dübendorf, Switzerland. [14]Department of Evolutionary Biology and Environmental Studies, University of Zurich, Winterthurerstrasse 190, 8057 Zürich, Switzerland. [15]Evolutionary Biology Group (@EvoHull), Department of Biological and Marine Sciences, University of Hull (UoH), Cottingham Road, Hull HU6 7RX, UK. [16]Institute of Biogeochemistry and Pollutant Dynamics (IBP), ETH Zurich, Zurich, Switzerland. [17]Department of Marine Sciences and Institute of Bioinformatics, University of Georgia, Georgia, USA. [18]These authors contributed equally: William Bernard Perry, Mathew Seymour. ✉e-mail: perryw1@cardiff.ac.uk; matsey@hku.hk; s.creer@bangor.ac.uk

