## [Peer Review File · Nature Communications]

An integrated spatio-temporal view of riverine biodiversity
using environmental DNA metabarcodingREVIEWER COMMENTS

Reviewer #1 (Remarks to the Author):

Overall this is a well written and presented manuscript that addresses the issue of consistency in biodiversity monitoring of riverine systems using eDNA. The results are noteworthy in that they do provide extensive data using intensive sampling across substantive spatiotemporal scale but do not reveal any particularly novel results compared to previous eDNA studies in riverine systems.

Two similar publications include;

Van Driessche et al (2023) Using environmental DNA metabarcoding to monitor fish communities in small rivers and large brooks: Insights on the spatial scale of information, *Environmental Research*, Volume 228.

Brys R et al (2021). Monitoring of spatiotemporal occupancy patterns of fish and amphibian species in a lentic aquatic system using environmental DNA. *Mol Ecol.* (13):3097-3110. doi: 10.1111/mec.15742.

The research is mostly sound and data has been presented and analysed appropriately. One suggestion in Figure 1 is to edit (b) so that there is clear demarcation between the three groups, perhaps a vertical line separating the group classes.

There is no indication in the sampling section of the Methods of what volume of water was sampled which is important for reproducibility.

One of the key concerns I had was regarding the transportation experiment using introduced extraneous DNA through dead Atlantic mackerel. There was no clear justification as to the choice of this approach. Were the fish freshly dead? Would there not have been variability in release of DNA during the decomposition process? It isn't clear that introducing dead and decaying carcasses is a good surrogate for eDNA production through living species as the type of eDNA being release would be different in origin. Life stage, biomass and spawning activity would have a greater influence on the variability of detection probability downstream. I am not convinced that this was the best experimental approach.

The overall results have provided further substantive data on the utility of eDNA as a tool for biodiversity monitoring in riverine systems. However, it falls short in providing a

definitive framework for how this monitoring tool could then be implemented as a consistent broadscale approach to whole ecosystem monitoring.

Reviewer #2 (Remarks to the Author):

General comments:

An integrated spatio-temporal view of riverine biodiversity using environmental DNA metabarcoding by Perry and Seymour et al. present an assessment of spatiotemporal variability across multiple river systems in the USA and Europe. This work reflects an intensive sampling strategy spanning two continents and 19 timepoints.

To better reflect this endeavour, I recommend the authors provide a bit more context in the Introduction and Results so readers are fully aware of the sampling setup/timepoints/number of rivers and locations. At present, the results are a little confusing to interpret without the full context/scope of the project, particularly given they precede the methods (as is this journal's style).

More significantly, I have reservations about the use of eDNA metabarcoding read counts as a proxy for abundance (used in this paper for Shannon index metrics and multivariate comparisons). Many metabarcoding papers report the effects of biomass, temperature, decay rate, collection/DNA extraction, PCR amplification biases (amongst other variables) on the downstream read counts in metabarcoding data, making it an inappropriate proxy for abundance metrics of individual species. The paper at present does not address these considerations nor justify their approach in using normalised (i.e., proportional) read count data for Shannon index metrics and multivariate comparisons between sites, rivers, and seasons. Seemingly, this abundance data is used for nearly all statistical analyses in this paper. As such, it is hard to interpret whether the spatio-temporal patterns reported by Perry and Seymour et al. in this paper reflect actual species variability in the sampled environment and are not just metabarcoding noise. I'd argue this needs to be addressed before publication.

Specific comments (based on line number in PDF):

Author contribution statement:

Line 35. From reading the author contribution statement, it seems that not all co-authors (n=22) contributed to the actual conception and/or undertaking of the project, just manuscript contributions. As outlined by the Nature editorial policies: "Each author is expected to have made substantial contributions to the conception or design of the work; or the acquisition, analysis, or interpretation of data; or the creation of new software used in the work; or have drafted the work or substantively revised it". Please assess whether all co-authors fulfil these criteria and if so, elaborate on contributions.

Abstract:

The abstract reads really well.

Line 52. Can you include the approximate extent of the sampled area, e.g., how many kilometres of river was sampled? This makes it easier to compare to other large-scale eDNA studies.

Line 52. Can you include the range of the timepoints (e.g., 19 timepoints across 2017 to 2018).

Introduction:

Line 88. Need space between 100 and km.

Paragraph 3 outlines the spatiotemporal question succinctly.

Line 94-97. This is a confusing sentence outlining the scope of the study sites and the eDNA metabarcoding aspect. Please re-word. You are assuming that readers understand how you are acquiring the "metazoan tree of life". Also please provide some details to the technical

replicates (e.g., X L samples) and 19 timepoints (e.g. across 2017-2018), particularly given the results will be provided before the methods.

Line 96. Is the longitudinal sampling of the three additional lake-fed rivers a separate component? As in they aren't sampled with the same temporal intensity as the River Conwy? Maybe write this in a separate sentence to the River Conwy sampling sentence, otherwise it is a bit confusing. Also please name these three lake-fed rivers.

Results:

Please provide some details regarding the sequencing statistics, i.e. how many total reads were sequenced across how many samples, what was the average read depth per sample per marker, were there any taxa detected in your laboratory controls? I also can't find the total number of taxa detected and how many of these were provided at species level.

Line 145. Again, please list the rivers. From the beginning of this manuscript, I don't think these rivers are listed anywhere except for in the Figure 3 caption.

Line 147. Distance from the lake or between sites? Please clarify all of these terms, particularly given the Results are preceding the Methods section.

Line 160-161. Its hard to see the driving pattern in the nMDS plots. It looks first and foremost that they are clustered by river, which you indicate has a significant impact on beta diversity (and presumably the largest effect), so maybe that is what the major colour scheme should be so it fits in with the above plots? I wonder if there is a better way to illustrate the effect of distance from the lake in the nMDS plots – from the plot itself this variable looks like its non-significant – but I think this is because the colour scheme is hard to separate – and obviously doesn't reflect your results reported on line 167. I will leave this to your judgement though.

Line 165-166. I think your figure references here are incorrect.

Line 178. What do you mean by samples across taxonomic groups? Do you mean the three markers showed the same dissimilarity patterns with distance down the river? Please re-word.

Line 180. Redefine nestedness in this context (i.e. compounding estimates of diversity with transport of eDNA downstream, or something similar). Interesting that the fish DNA is nested, whilst the metazoans and arthropods or not. Is this based on 'quantitative' proportional abundance data or presence-absence?

Line 204. How many seasons and in what year were you able to sample across? Again, the temporal extent of sampling is not clear and not elaborated on in the Methods (nor Results!).

Discussion:

Line 256. You previously report nestedness of the 12S fish data down the rivers – would this not explain the increase in fish alpha diversity downstream?

Line 293-294. You report that eDNA transport down the River Conwy is 5km. How far apart are your sites down this river? Would this not affect nestedness?

Line 300. Presuming your relative abundance metrics are accurate and not affected by primer amplification biases or template complexity – see my comments on Methods.

Conclusion:

Line 371. Was there not a confounding (nestedness) effect on fish communities in the River Conwy?

Line 372. Report the transportation (i.e., 5km) you found with your mackerel experiment.

Line 374: Emphasise cross-river comparisons across northern hemisphere continents.

Methods:

Lines 384-393. Please provide details of sampling months and year sampled? How many kilometres of each river was sampled? How many litres of water were collected per sample? Reiterate how many technical replicates were collected at each site, as this is not clear. How many samples were collected in total? What were the water samples filtered on (size of pore?), using what filtration system? Did you include any filtration controls in your sampling to assess cross-contamination of equipment? This sampling paragraph is very light on information and does not provide a good reflection of all the hard sampling work that would have been undertaken.

Line 385. Please expand on this River Continuum Concept approach.

Line 409. Does the two-step protocol include dual indexing in both rounds? Please elaborate.

Line 410. Triplicate eDNA samples per site? How many PCR replicates per sample?

Line 412. Missing "to" after prior.

Line 413 and 417. PCR2 amplicons? Are you referring to amplicons from the second-step PCR? Please re-write.

Line 421. Do you mean 100,000 reads per amplicon sample?

Line 421-423. Were negative controls spiked into the final library for sequencing? What contamination-control practices were implemented? Was the lab work conducted in a clean, pre-PCR facility?

Line 448. Why a query coverage of 90%? That seems quite low when trying to obtain an accurate species ID. Surely a species ID would warrant 100% query coverage with a high percent identity match – particularly when the fragment sizes in question are quite small.

How can you be sure that your species ID is correct (particularly between closely-related species) when you are effectively allowing 10% of your sequence to not be matched to the reference sequence? In addition, can you provide a table of your ASVs and their corresponding % identity match and % query coverage.

Line 475. Why was rarefaction not implemented? Surely if you are comparing between sites and timepoints you would need the same number of reads across samples so as not to inflate diversity estimates. Please provide species accumulation curves by sequence depth for each site/timepoint (i.e. the combined triplicate since they are being merged) to assess whether accumulation has plateaued and therefore whether rarefaction is required or not.

Line 477. Can you explain how your normalised read counts are an appropriate proxy for abundance data? Would the read count per species not be affected by primer amplification biases (particularly with a two-step PCR) and sequencing depth? Many metabarcoding studies have reported that the observed proportion of species from eDNA metabarcoding data (in contrast to targeted species-specific assays) do not reflect the true proportion in the environment. For example, this recent study -Shelton et al. 2022; <https://esajournals.onlinelibrary.wiley.com/doi/full/10.1002/ecy.3906> found that “that simple tabulations of proportions of amplicon-sequence reads are likely to provide misleading inferences due amplification bias”. The above authors needed to calibrate the data to yield estimates of proportional contributions of each of the taxa that reflect the original biological sample prior to PCR. Recently, Gold et al. 2023 (<https://journals.plos.org/plosone/article?id=10.1371/journal.pone.0285674>) reported that “metabarcoding datasets are strongly affected by (1) deterministic amplification biases during PCR and (2) stochastic sampling of amplicons during sequencing—both of which we can model—but also by (3) stochastic sampling of rare molecules prior to PCR, which remains a frontier for quantitative metabarcoding”.

Line 480. Please explain how you can use quantitative abundance data from eDNA metabarcoding for Shannon index values? Also why did you choose not to extend this to the 12S fish ASVs?

Line 504. Is the multivariate analysis (i.e. Bray-Curtis mMDS plots) based on normalised read count data for all three markers, or is the 12S data being treated different as it was for the alpha diversity analysis? Again, I am concerned about the quantitative aspect whereby you are inputting uncalibrated metabarcoding data as a proxy of varying abundance between sites. Are you seeing the same site variation patterns (i.e. distance from the lake) if you convert your data to presence-absence and conduct Jaccard nMDS plots?

REVIEWER COMMENTS

Reviewer #1 (Remarks to the Author):

Overall this is a well written and presented manuscript that addresses the issue of consistency in biodiversity monitoring of riverine systems using eDNA. The results are noteworthy in that they do provide extensive data using intensive sampling across substantive spatiotemporal scale but do not reveal any particularly novel results compared to previous eDNA studies in riverine systems. Two similar publications include;

Van Driessche et al (2023) Using environmental DNA metabarcoding to monitor fish communities in small rivers and large brooks: Insights on the spatial scale of information, Environmental Research, Volume 228.

Brys R et al (2021). Monitoring of spatiotemporal occupancy patterns of fish and amphibian species in a lentic aquatic system using environmental DNA. Mol Ecol. (13):3097-3110. doi: 10.1111/mec.15742.

We thank reviewer 1 for their appreciation in seeing the value of our study and for drawing attention to these two studies, however, despite the common theme of eDNA, we would argue that our study is substantially different:

- 1) The highlighted studies mainly focus on fish, whereas we show results from multiple taxa, as well as a multi-marker approach, allowing us to assess the biodiversity within the freshwater ecosystem in unprecedented depth.
- 2) Both studies focus on methodology, assessing and confirming the efficacy of eDNA approaches in lentic and small rivers and brooks, whereas we are applying the proven methodology to model riverine biodiversity.
- 3) The spatial scales in these two studies highlighted are based on one location, and 50m (Brys R et al 2021) and 3km reaches (Van Driessche et al 2023), whereas we massively expand on this, showing results from five international rivers and on the scale of 30kms (i.e. whole river catchment scale).
- 4) In our study, we focus on communities shifts across a whole year, whereas the suggested studies do not include temporal replication or temporal sampling is on the scale of weeks and so does not assess the diversity across seasons.
- 5) The highlighted studies focus entirely on caged animals, and not on natural communities, unlike our study. Here we have used natural communities to better understand the spatio-temporal resolution of biodiversity measures which eDNA analysis can provide, in addition to using artificial communities to measure eDNA transport through the mackerel experiment.
- 6) the study by Brys R et al. (2021) is a study on lentic communities, a completely different habitat to that examined in our study.

By combining space, time, environmental conditions and biodiversity targets across the tree of life, we believe that our study presents originality that will be highly valued by the research community.

The research is mostly sound and data has been presented and analysed appropriately. One suggestion in Figure 1 is to edit (b) so that there is clear demarcation between the three groups, perhaps a vertical line separating the group classes.

By “three groups”, we believe Reviewer #1 is referring here to the three sections of the river. We have now included vertical lines to separate these sections in figure 1.

There is no indication in the sampling section of the Methods of what volume of water was sampled which is important for reproducibility.

We thank the reviewer for pointing out this oversight and we have added the information:

Line 98-99: "...each with three 1L sample replicates."

Line 409: "Three replicate water samples (1L per sample) were collected at each sample site and time point."

One of the key concerns I had was regarding the transportation experiment using introduced extraneous DNA through dead Atlantic mackerel. There was no clear justification as to the choice of this approach. Were the fish freshly dead? Would there not have been variability in release of DNA during the decomposition process? It isn't clear that introducing dead and decaying carcasses is a good surrogate for eDNA production through living species as the type of eDNA being release would be different in origin. Life stage, biomass and spawning activity would have a greater influence on the variability of detection probability downstream. I am not convinced that this was the best experimental approach.

While we agree that this experimental approach could be further improved, the scale of the catchment (30,000m+) means that using dead tissue gives an approximation of how far the eDNA is likely to travel. The aim of the study is to examine natural communities, and the release of natural mackerel DNA was a small element of the overall study design, and a proof of concept. The addition of this element adds a further complexity and allows the exploration of DNA transport in a natural ecosystem. Unlike previous experiments which have introduced caged invasive species to lentic waterbodies, here we are examining river systems and the potential for an introduction of an invasive species must therefore be prevented at all times, hence the use of freshly killed fish samples.

We have added the following text to the methods:

Line 432-434: "To introduce a simple and traceable exogenous source of eDNA without risks associated with the introduction of non-native species, five dead fresh Atlantic mackerel (*Scomber scombrus*) (~1.5kg at any one time) were placed in a plastic mesh bag at the head of the River Conwy..."

In addition to this, there would be welfare issues of maintaining living non-native species in this environment for an annual period.

The overall results have provided further substantive data on the utility of eDNA as a tool for biodiversity monitoring in riverine systems.

However, it falls short in providing a definitive framework for how this monitoring tool could then be implemented as a consistent broadscale approach to whole ecosystem monitoring.

We thank the reviewer for this overall positive appraisal and we agree that this is an important message to get across. We have therefore added additional text to the conclusion to reflect this:

Line 402-404: "We would therefore encourage progressive dialogue between researchers and stakeholders to enhance the standardization of eDNA metabarcoding approaches for whole ecosystem biodiversity assessment."

Reviewer #2 (Remarks to the Author):

General comments:

An integrated spatio-temporal view of riverine biodiversity using environmental DNA metabarcoding by Perry and Seymour et al. present an assessment of spatiotemporal variability across multiple river systems in the USA and Europe. This work reflects an intensive sampling strategy spanning two continents and 19 timepoints.

To better reflect this endeavour, I recommend the authors provide a bit more context in the Introduction and Results so readers are fully aware of the sampling setup/timepoints/number of rivers and locations. At present, the results are a little confusing to interpret without the full context/scope of the project, particularly given they precede the methods (as is this journal's style).

We thank Reviewer #2 for their comments and have now included further details regarding the sampling effort at the beginning of the Results section, where there is less of a word limitation, which now reads:

Line 109-113: "A total of 798 water samples were taken from 14 sites and 19 timepoints (27th April 2017 to 18th April 2018) along the River Conwy (Wales, UK). In addition to these samples, in July 2017, 39 samples were collected along the River Glatt (Switzerland), 33 samples were collected along the River Gwash, (England, UK), 36 samples were collected along the River Tywi (Wales, UK) and 33 samples were collected along the Skaneateles Creek (USA)."

While we agree greater information on the sampling would be valuable in the introduction, we are heavily restricted in terms of word count (500 words). The number of sites, timepoints, number of rivers and locations is already presented at the end of the introduction, but, guided by Reviewer #2's comments, we have added more detail, and it now reads:

Line 97-100: "Here, we intensively sampled 14 sites along the River Conwy, a well-documented lake-fed river in Wales (UK)¹⁴, at 19 timepoints over a year (April 2017 to April 2018), each with three 1L sample replicates. Sampling of three additional lake-fed rivers in Europe (Tywi, Gwash and Glatt) and one in North America (Skaneateles Creek) was carried out at one timepoint."

More significantly, I have reservations about the use of eDNA metabarcoding read counts as a proxy for abundance (used in this paper for Shannon index metrics and multivariate comparisons). Many metabarcoding papers report the effects of biomass, temperature, decay rate, collection/DNA extraction, PCR amplification biases (amongst other variables) on the downstream read counts in metabarcoding data, making it an inappropriate proxy for abundance metrics of individual species. The paper at present does not address these considerations nor justify their approach in using normalised (i.e., proportional) read count data for Shannon index metrics and multivariate comparisons between sites, rivers, and seasons. Seemingly, this abundance data is used for nearly all statistical analyses in this paper. As such, it is hard to interpret whether the spatio-temporal patterns reported by Perry and Seymour et al. in this paper reflect actual species variability in the sampled environment and are not just metabarcoding noise. I'd argue this needs to be addressed before publication.

We have improved the clarity of the methods to state more explicitly where we have used abundance and where we have used presence/absence:

Line 549-551: “The linear model had the response variable: species count (not based on normalised reads, instead based on presence or absence) from 12S fish ASVs...”

Line 575-577: “Beta diversity in the form of Sørensen dissimilarity was also calculated and further partitioned into nestedness and species replacement (turnover), derived using the ‘betapart’ package 69, and was based on presence and absence.”

Line 545-548: “‘Vegan’ v 2.4.2 ⁶⁷ was used to calculate Shannon index values for the 18S and COI markers using the normalised read counts for the proportion an ASV contributes to a community and number of ASVs as a proxy for species number.”

We used abundances for all 18S and COI analyses, but not the 12S where species count was used instead (presence/absence). Shannon index was used for the COI and 18S as the disparity in reads between ASVs was more marked, along with the samples having much greater diversity, and therefore information on which ASVs were rare or common in a sample, as provided by the Shannon index, was useful information.

We are confident that the abundances used in this study for 18S and COI are informative of ecological patterns, and not patterns of metabarcoding noise because of the following reasons:

1. In a companion, single species study derived from the present Conwy dataset and from the same sequencing runs, we found significant correlations between Atlantic salmon abundances and rod catch data (a proxy for biomass), providing evidence that the abundance from these metabarcoding runs are meaningful. The current data analyses are very expansive, focused on multi-species biodiversity and space is tight and so we are choosing to present the single species findings elsewhere.
2. The mackerel experiment additionally illustrates that relative abundance data provides spatial quantitative information about how much eDNA from a species is in an area. Higher abundances mean more eDNA and that you are closer to the source of the eDNA:

Supplementary figure 1: Read proportions from the introduced dead Atlantic mackerel (*Scomber scombrus*) at the source of the River Conwy: Llyn Cowy (i.e. 0m distance from the lake). Plots are for reads produced using the (a) 12S and (b) COI marker. Blue line is a fit loess smoothing line.

Therefore, incorporating abundance, and thus information on proximity, is important for calculating diversity metrics, as low read abundance species are either at low biomass, or are further away, and not contributing as much to the community in the part of the river being sampled. Treating low abundance species the same as high abundance species through presence/absence therefore skews the observer's perspective of the community in a location.

The additional mackerel plot has now been added to the supplementary materials (supplementary figure 1).

3. The modelling included in this study, particularly the GAMs, should be robust to noise introduced if read abundances does not always reflect actual abundance.
4. If abundance was largely noise introduced by the metabarcoding pipeline, and not based on the biological communities, we would expect that noise to be consistent across the entire dataset. Yet, we detect different spatiotemporal/environmental responses from different taxonomic groups.
5. We have reanalysed the entire Conwy dataset using presence/absence data to demonstrate the utility of using abundance data, and demonstrate that most of the trends are robust across the different data types:

12S, 18S and COI beta diversity

There were minimal changes seen. For example, the 18S PERMANOVA, out of the 57 combinations of phyla and variables tested, there were only four discrepancies (7%) in significance between the analysis featuring abundance and presence/absence data. Using abundance found significant impacts of environmental variables where presence/absence did not. Although the p values for presence/absence in these cases were not significant, they were not far from it (p value range of those that became insignificant in presence/absence analysis: 0.061 – 0.191). A full model output for presence/absence and abundance is provided.

For COI PERMANOVA, there were no discrepancies in significance between the analysis featuring abundance and presence/absence data.

For the 12S PERMANOVA, out of the 11 variables tested, there were only two discrepancies (18%) in significance between the analysis featuring abundance and presence/absence data. As for the 18S, using abundance found significant impacts of environmental variables where presence/absence did not. A full model output for presence/absence and abundance is provided.

NMDS plot 18S:

Similar trend in terms of importance distance from the lake plays, but it's flipped on the x axis.

NMDS plot COI:

Similar trend, again.

NMDS plot 12S:

Abundance used (Bray)

Presence/absence used (Jaccard)

Using the presence absence loses granularity, with samples suddenly all having the same number of species, as it is not accounting for species that are at low abundance. However, overall, the trend remains the same.

18S and COI alpha diversity

To test the impact of swapping abundance with presence/absence for alpha diversity, instead of using the Shannon index as a measure of alpha diversity, which requires abundances to calculate evenness (the distribution of abundances among species), it was instead replaced with species richness (number of species). Spatiotemporal trends were robust between alpha diversity metrics (abundance vs presence/absence) with few exceptions (e.g., the presence/absence meant the winter diversity signal was exaggerated in nematodes, and flattened in molluscs):

18S plot:

Abundance used

Presence/absence used

COI plot:

Considering other parameters, a larger impact of using presence/absence was seen on alpha diversity when compared to the change seen on beta diversity. Out of the 63 combinations of phyla and variables tested, there were 29 discrepancies (38%) in significance between the analysis featuring abundance and presence/absence data. 20 of these discrepancies occurred as the result of the presence/absence changing non-significant results to significant. This shift is likely due to very rare species now having an equal weighting to common species. We predict that the removal of information on abundance, and thus evenness, means that the effect size of the variables is disproportionately amplified, thus generating more significant effects.

Author contribution statement:

Line 35. From reading the author contribution statement, it seems that not all co-authors (n=22) contributed to the actual conception and/or undertaking of the project, just manuscript contributions. As outlined by the Nature editorial policies: "Each author is expected to have made substantial contributions to the conception or design of the work; or the acquisition, analysis, or interpretation of data; or the creation of new software used in the work; or have drafted the work or substantively revised it". Please assess whether all co-authors fulfil these criteria and if so, elaborate on contributions.

We thank the reviewer for highlighting this oversight and we have updated the authorship statement with the correct contributions and would like to reassure the reviewer that all co-authors fulfil the journal requirements to be named as such on this manuscript:

Line 36-46: "K.D., M.de B., M.S, N.M., F.E., K.W., B.E., H.M.B., J.C., B.J.C., I.D. and S.C. conceived and designed the study. The manuscript was written by W.B.P, with contributions from all co-authors (M.S., L.O., I.B.J., N.M., F.E., R.H., M. de B., I.B., K.W., B.E., R.B., F.A., R.D., L.L.H., E.M., K.D., H.M.B., G.C., J.C., B.J.C., I.D. and S.C.). M.S., R.H., R.B., I.B., B.E, F.A., L.L.H, K.D., G.C. and E.M. were involved in sample collection and the acquisition of data. W.B.P, K.D., M.de B., L.O., N.M., F.E., I.B.J., H.M.B., J.C., B.J.C., G.C., I.D. and S.C. and were involved in the interpretation of data. M.S. contributed to field sampling design and co-ordinated field and laboratory operations, including DNA extraction and sequencing library preparation in collaboration with I.B.. L.O. further contributed to sequencing library preparation and sequencing. Bioinformatics and statistical analyses were carried out by W.B.P with

E.M. contributing to analyses on nestedness and turnover. Meteorological and hydrological metadata on the River Conwy associated with the eDNA samples was provided by B.J.C.”

Abstract:

The abstract reads really well.

We thank the Reviewer #2 for their positivity.

Line 52. Can you include the approximate extent of the sampled area, e.g., how many kilometres of river was sampled? This makes it easier to compare to other large-scale eDNA studies.

This has now been included, and reads:

Line 56-57: “Here, using intensive, spatio-temporal eDNA sampling across space (five rivers in Europe and North America, with an upper range of 20-35 km between samples)...”

Due to the tight word count, words have been cut elsewhere in the abstract.

Line 52. Can you include the range of the timepoints (e.g., 19 timepoints across 2017 to 2018).

Changed and now reads:

Line 57-58: “...time (19 timepoints between 2017 and 2018)...”

Introduction:

Line 88. Need space between 100 and km.

Changed and now reads:

Line 92: “...100 km...”

Paragraph 3 outlines the spatiotemporal question succinctly.

We thank the reviewer for their comment.

Line 94-97. This is a confusing sentence outlining the scope of the study sites and the eDNA metabarcoding aspect. Please re-word. You are assuming that readers understand how you are acquiring the “metazoan tree of life”.

We thank the reviewer for highlighting this and have clarified the text, it now reads:

Line 100-102: “Diversity was assessed across the metazoan tree of life using three genetic markers, each offering identification of different taxa.”

Also please provide some details to the technical replicates (e.g., X L samples) and 19 timepoints (e.g. across 2017-2018), particularly given the results will be provided before the methods.

These details have now been added, and it now reads:

Line 97-99: “Here, we intensively sampled 14 sites along the River Conwy, a well-documented lake-fed river in Wales (UK)¹⁴, at 19 timepoints over a year (April 2017 to April 2018), each with three 1L sample replicates.”

Line 109-113: "A total of 798 water samples were taken from 14 sites and 19 timepoints (27th April 2017 to 18th April 2018) along the River Conwy (Wales, UK). In addition to these samples, in July 2017, 39 samples were collected along the River Glatt (Switzerland), 33 samples were collected along the River Gwash, (England, UK), 36 samples were collected along the River Tywi (Wales, UK) and 33 samples were collected along the Skaneateles Creek (USA)."

Line 96. Is the longitudinal sampling of the three additional lake-fed rivers a separate component? As in they aren't sampled with the same temporal intensity as the River Conwy? Maybe write this in a separate sentence to the River Conwy sampling sentence, otherwise it is a bit confusing. Also please name these three lake-fed rivers.

We thank the reviewer for highlighting this and have clarified the text. We have separated the two sentences, and elaborated that the other lake fed rivers were only sampled at one timepoint. The names of the three lake-fed rivers have been added, and now reads:

Line 109-113: "A total of 798 water samples were taken from 14 sites and 19 timepoints (27th April 2017 to 18th April 2018) along the River Conwy (Wales, UK). In addition to these samples, in July 2017, 39 samples were collected along the River Glatt (Switzerland), 33 samples were collected along the River Gwash, (England, UK), 36 samples were collected along the River Tywi (Wales, UK) and 33 samples were collected along the Skaneateles Creek (USA)."

Results:

Please provide some details regarding the sequencing statistics, i.e. how many total reads were sequenced across how many samples, what was the average read depth per sample per marker

We have now added information on the read depth per sample and marker to the result section, and reads:

Line 113-115: "A total of 896 samples were successfully sequenced, producing 178,833,278 12S (average of 199,591 per sample), 144,016,790 18S (average of 160,733 per sample) and 279,175,484 COI (average of 311,580 per sample) reads."

Were there any taxa detected in your laboratory controls?

Largely, no, but there were a few exceptions. We have now added this text to add more information to the methods:

Line 475-480: "Of the 134 field and laboratory negative controls, the majority failed to produce any reads or pass the DADA read quality filtering, read merging and downstream filtering of merged reads. 126 (94%), 92 (69%) and 119 (89%) of the negative controls failed quality control for the 12S, 18S and COI, respectively. Of the remaining reads detected in the negative controls, these were used to filter reads associated with samples using the R package microDecon 1.0.2⁵⁵, except for 12S, where the limited number of taxonomic groups meant that the approach was not suitable."

I also can't find the total number of taxa detected and how many of these were provided at species level.

We did not concentrate on describing taxonomy to the species level. Instead, we assigned taxonomy to phyla and then used ASVs. Information on ASVs for each of the markers can be found in supplementary table 5.

Line 145. Again, please list the rivers. From the beginning of this manuscript, I don't think these rivers are listed anywhere except for in the Figure 3 caption.

The names of the rivers originally and still feature in the Methods section, but we have now also added them into Introduction and Results sections for clarity, which now reads:

Line 97-100: "Here, we intensively sampled 14 sites along the River Conwy, a well-documented lake-fed river in Wales (UK)¹⁴, at 19 timepoints over a year (April 2017 to April 2018), each with three 1L sample replicates. Sampling of three additional lake-fed rivers in Europe (Tywi, Gwash and Glatt) and one in North America (Skaneateles Creek) was carried out at one timepoint."

Line 109-113: "A total of 798 water samples were taken from 14 sites and 19 timepoints (27th April 2017 to 18th April 2018) along the River Conwy (Wales, UK). In addition to these samples, in July 2017, 39 samples were collected along the River Glatt (Switzerland), 33 samples were collected along the River Gwash, (England, UK), 36 samples were collected along the River Tywi (Wales, UK) and 33 samples were collected along the Skaneateles Creek (USA)."

Line 147. Distance from the lake or between sites? Please clarify all of these terms, particularly given the Results are preceding the Methods section.

Apologies for this ambiguity. We have now explicitly stated throughout the results "distance from the lake" rather than just "distance", and have also specified when we are talking about distance between samples (i.e. for the nestedness and turnover analysis).

Line 160-161. Its hard to see the driving pattern in the nMDS plots. It looks first and foremost that they are clustered by river, which you indicate has a significant impact on beta diversity (and presumably the largest effect), so maybe that is what the major colour scheme should be so it fits in with the above plots? I wonder if there is a better way to illustrate the effect of distance from the lake in the nMDS plots – from the plot itself this variable looks like its non-significant – but I think this is because the colour scheme is hard to separate – and obviously doesn't reflect your results reported on line 167. I will leave this to your judgement though.

Reviewer #2 is quite right, the main driver of beta diversity is the river being sampled, which we hope is clear from the red epicentres of each cluster in figure 3e-g, as well as the use of different shapes for each river, depicted in panel figure 3a. There is a lot to display on one plot, but by zooming into each cluster for each river, the pale blues are separated from the dark blues, demonstrating the significant impact of distance from the lake.

Line 165-166. I think your figure references here are incorrect.

We thank the reviewer for pointing this out. This has now been rectified.

Line 178. What do you mean by samples across taxonomic groups? Do you mean the three markers showed the same dissimilarity patterns with distance down the river? Please re-word.

We apologies for the poorly worded sentence and have changed the sentence to clarify this. It now reads:

Line 197-200: "In the Conwy, samples collected spatially close together had lower dissimilarity than samples collected further apart, with dissimilarity increasing with distance between samples, and this

was seen across each of the taxonomic groups (figure 5a,b,c), indicating that communities change along the course of the river.”

Line 180. Redefine nestedness in this context (i.e. compounding estimates of diversity with transport of eDNA downstream, or something similar).

The argument that we make in the discussion is that the observed nestedness is not because of transport, and that the nested diversity is driven by the ecology of the organisms.

Interesting that the fish DNA is nested, whilst the metazoans and arthropods or not. Is this based on ‘quantitative’ proportional abundance data or presence-absence?

Nestedness for fish, metazoans and arthropods was calculated based on presence/absence. We have clarified this in the methods, which now reads:

Line 575-577: “Beta diversity in the form of Sørensen dissimilarity was also calculated and further partitioned into nestedness and species replacement (turnover), derived using the ‘betapart’ package⁶⁹, and was based on presence and absence.”

As we outline in the discussion:

Line 277-280: “ The increase in fish alpha diversity downstream (figure 2d) reflects known distribution patterns of freshwater fish communities^{22–25} and their propensity to be nested^{26,27}, due to increased availability, size and heterogeneity of habitats²⁸ coupled with increased accessibility to diadromous and potadromous species.”

This may not be the case for metazoans and arthropods (we could not find any literature on it being measured previously in these taxonomic groups).

Line 204. How many seasons and in what year were you able to sample across? Again, the temporal extent of sampling is not clear and not elaborated on in the Methods (nor Results!).

This information has now been included at the start of the Results section (Line 104) and the Methods section, and now reads:

Line 97-98: “Here, we intensively sampled 14 sites along the River Conwy, a well-documented lake-fed river in Wales (UK)¹⁴, at 19 timepoints over a year (April 2017 to April 2018)...”

Line 109-113: “A total of 798 water samples were taken from 14 sites and 19 timepoints (27th April 2017 to 18th April 2018) along the River Conwy (Wales, UK). In addition to these samples, in July 2017, 39 samples were collected along the River Glatt (Switzerland), 33 samples were collected along the River Gwash, (England, UK), 36 samples were collected along the River Tywi (Wales, UK) and 33 samples were collected along the Skaneateles Creek (USA).”

Discussion:

Line 256. You previously report nestedness of the 12S fish data down the rivers – would this not explain the increase in fish alpha diversity downstream?

Yes, absolutely. We comment on the documented nested nature of fish communities in the last paragraph of this section, but we have now also added it into the text in this second paragraph:

Line 277-280: “ The increase in fish alpha diversity downstream (figure 2d) reflects known distribution patterns of freshwater fish communities^{22–25} and their propensity to be nested^{26,27}, due to increased availability, size and heterogeneity of habitats²⁸ coupled with increased accessibility to diadromous and potadromous species.”

Line 293-294. You report that eDNA transport down the River Conwy is 5km. How far apart are your sites down this river? Would this not affect nestedness?

The eDNA transport downstream was a maximum of 5kms, but the majority of the time it was not detected over 1km. We realise this was missing from the text and so have added clarification:

Line 117-121: “The Atlantic mackerel (*Scomber scombrus*), used as an introduced source of eDNA at the outlet of Llyn Conwy (the source of the River Conwy), was detected at sites E01, E02, E03, E04 and E05 (figure 1a) a total of 30 times across the study with the 12S marker, reaching a maximum transport of 5,000 m downstream of the release site. Although, in most cases it was not detected beyond 1,000 m downstream (Supplementary figure 1).”

Addition of Supplementary figure 1 helps to better explain the transport.

As displayed in figure 1, as well as on the x axis of figure 2, 3 and supplementary figure 3, samples down the River Conwy were taken across a 35 km range. The average distance between most sites being 3.4 kms. Therefore, it is possible that transport could have some role in describing the nestedness. However, we believe that this is not the case, for the following reasons:

1. Although maximum transport detected was 5kms, most of the time the mackerel was not detected beyond 1km.
2. Even considering the upper transport value of 5kms, we would expect eDNA to be transported downstream to the next site over, but no further than that, because of the distances between samples sites.
3. We know that fish communities have a propensity to be nested, with increasing alpha diversity moving downstream.
4. If transport was the main driver behind the nestedness seen in fish, we would expect to see it as a driver of communities detected by the 18S and COI markers, but these were characterized by turnover.

Line 300. Presuming your relative abundance metrics are accurate and not affected by primer amplification biases or template complexity – see my comments on Methods.

We have responded to this concern within the ‘general comments’ section.

Conclusion:

Line 371. Was there not a confounding (nestedness) effect on fish communities in the River Conwy?

Due to the relatively short transport of eDNA (max of 5kms, but mainly 1km, from mackerel experiment) and large scale of the sampling (35kms of river), transport could not have produced the strong nested patterns we observed. We also do not see any nestedness in the other markers, demonstrating that they are not being dictated by eDNA transport.

Instead, 12S results are demonstrating the naturally occurring, documented, nested fish community structure seen in other rivers.

Line 372. Report the transportation (i.e., 5km) you found with your mackerel experiment.

As requested, we have added clarification with the following text:

Line 394-397: “The relatively short transport distance of eDNA (i.e. a maximum of 5,000 m in the mackerel experiment) means that detectable diversity patterns, especially using 12S and 18S markers, are congruent with established ecological trends yielded via conventional, but more costly, low throughput non-molecular approaches.”

Line 374: Emphasise cross-river comparisons across northern hemisphere continents.

As requested, we have added emphasis with the following text:

Line 397-399: “Cross-river comparisons among rivers from Europe and North America showed that trends in alpha and beta diversity along the river were largely consistent, and in accordance with the ecological characteristics of those rivers.”

Methods:

Lines 384-393. Please provide details of sampling months and year sampled?

We have now added the sample dates for each of the rivers, which now reads:

Line 410-416: “Sample sites were arranged in a linear longitudinal transect along the River Conwy (14 sample sites, over a 35.2 km stretch of river in Wales (UK), sampled between 27th April 2017 to 18th April 2018) (figure 1a), River Tywi (12 sample sites, over a 25.7 km stretch of river in Wales (UK), sampled on 13th July 2017), and River Gwash (11 sample sites, over a 27.4 km stretch of river in England (UK), sampled on 31st July 2017), River Glatt (13 sample sites, over a 35.1 km stretch of river in Switzerland, sampled on 3rd July 2017) and Skaneateles Creek, USA (11 sample sites, over a 20 km stretch of river in the USA, sampled on 19th July 2017)(supplementary table 1).”

How many kilometres of each river was sampled?

We have added the length of the stretch of river as requested, which now reads:

Line 410-416: “Sample sites were arranged in a linear longitudinal transect along the River Conwy (14 sample sites, over a 35.2 km stretch of river in Wales (UK), sampled between 27th April 2017 to 18th April 2018) (figure 1a), River Tywi (12 sample sites, over a 25.7 km stretch of river in Wales (UK), sampled on 13th July 2017), and River Gwash (11 sample sites, over a 27.4 km stretch of river in England (UK), sampled on 31st July 2017), River Glatt (13 sample sites, over a 35.1 km stretch of river in Switzerland, sampled on 3rd July 2017) and Skaneateles Creek, USA (11 sample sites, over a 20 km stretch of river in the USA, sampled on 19th July 2017)(supplementary table 1).”

How many litres of water were collected per sample?

We have added the volume of water sampled to the text, which now reads:

Line 409: “Three replicate water samples (1L per sample) were collected at each sample site and time point.”

Reiterate how many technical replicates were collected at each site, as this is not clear.

We have clarified the text as requested. Three replicates were taken, but given that we are sampling in lotic systems, we would consider the replicates as ecological replicates, rather than technical; to avoid any confusion, we have simply termed them as 'replicates'. The text has now been added to the beginning of the 'sampling' section. It now reads:

Line 409: "Three replicate water samples (1L per sample) were collected at each sample site and time point."

How many samples were collected in total?

939 samples were collected. This information has also been added to the 'sampling section', which now reads:

Line 417: "A total of 939 samples were taken."

What were the water samples filtered on (size of pore?), using what filtration system?

We have added the following details to the text:

Line 422-423: "Water samples were filtered through 0.22 μm SterivexTM filter units (EMD Millipore Corporation) using a Geopump TM Series II peristaltic pump (Geotech)."

Did you include any filtration controls in your sampling to assess cross-contamination of equipment?

Yes, negative controls in the field were taken to assess cross contamination. This information has now been included in the sampling section, which now reads:

Line 423-425: "In the field, 66 negative controls were taken using deionized water and were treated the same as the other samples through downstream processing. Also processed and sequenced were 68 laboratory negative controls."

This sampling paragraph is very light on information and does not provide a good reflection of all the hard sampling work that would have been undertaken.

We apologies for this oversight. We have added additional detail to this section and we hope that this has now been rectified with the addition of the requests above.

Line 385. Please expand on this River Continuum Concept approach.

This phrase complicates the methodology, and so we have changed the sentence to:

Line 410: "Sample sites were arranged in a linear longitudinal transect..."

Line 409. Does the two-step protocol include dual indexing in both rounds? Please elaborate.

We have clarified the protocol within the step. Dual indexing is added in the second round of PCR. This information has now been added and now reads:

Line 448-450: "A two-step library preparation method was used following Bista et al. (2017), but employing 4 sets of unique dual indexed 96 tags (n=384) in the second round of PCR to facilitate multiplexing and to reduce cross-talk between samples in downstream analyses as in Brennan et al. (2019)."

Line 410. Triplicate eDNA samples per site? How many PCR replicates per sample?

There were three water samples taken at every sample site, at each time point, which we have now clarified in the text, with the text now reading:

Line 409: "Three replicate water samples (1L per sample) were collected at each sample site and time point."

There were also three PCR replicates for each of those samples. It now reads:

Line 450-452: "First round PCR was done in triplicate for every sample and each of the three PCR primers, using Q5 HS High-Fidelity mastermix (New England Biolabs) for the 12S and 18S markers..."

Line 454: "Triplicates were pooled and underwent second round PCR to add unique dual indexes."

Line 412. Missing "to" after prior.

Sentence removed during revisions.

Line 413 and 417. PCR2 amplicons? Are you referring to amplicons from the second-step PCR? Please re-write.

We have clarified the text with the following:

Line 454-457: "The second round PCR used Q5 HS High Fidelity Master Mix and amplicons from the second round PCR were purified twice using AMPure magnetic beads and quantitated using a 200 plate reader (TECAN) using qubit dsDNA HS solution (Invitrogen)."

Line 421. Do you mean 100,000 reads per amplicon sample?

We meant 100,000 reads per sample for each of the genes. We have tried to clarify the text, which now reads:

Line 460-462: "The final molarity of the pools was confirmed using a HS D1000 tapestation screentape (Agilent) prior to 250 bp paired-end sequencing on an Illumina HiSeq platform aiming for 100,000 reads per sample and target gene (e.g. 12S, 18S and COI)."

Line 421-423. Were negative controls spiked into the final library for sequencing?

In addition to the 66 negative controls taken in the field, 68 laboratory negative controls were also processed and sequenced. This text has been added to the 'DNA extraction and library preparation' section and now reads:

Line 423-425: "In the field, 66 negative controls were taken using deionized water and were treated the same as the other samples through downstream processing. Also processed and sequenced were 68 laboratory negative controls."

What contamination-control practices were implemented? Was the lab work conducted in a clean, pre-PCR facility?

We have now added the additional text to the 'DNA extraction and library preparation' section:

Line 426-429: "All pre-PCR steps were performed in a PCR free, eDNA clean room, in a separate building to where the PCRs were undertaken. Access to the clean room is restricted to trained users and the laboratory is regularly cleaned with bleach. Those using the PCR free room wear PCR free overcoats, hair nets, shoes, gloves and masks."

Line 448. Why a query coverage of 90%? That seems quite low when trying to obtain an accurate species ID. Surely a species ID would warrant 100% query coverage with a high percent identity match – particularly when the fragment sizes in question are quite small. How can you be sure that your species ID is correct (particularly between closely-related species) when you are effectively allowing 10% of your sequence to not be matched to the reference sequence?

All analyses were conducted on ASVs, and so as long as those ASVs could be assigned to the phyla level, precise species identification was superfluous. Therefore, because we only needed accurate identification to the phyla level, a slightly more conservative query cover was appropriate. We took this approach because of the problems with assigning taxonomy to the species level, due to incomplete reference databases. The exception to this was the 12S region, which, because of its specificity to fish, allowed higher taxonomic resolution, and thus identification to the species level, with species number being used for downstream analysis rather than ASVs.

In addition, can you provide a table of your ASVs and their corresponding % identity match and % query coverage.

We have now provided reviewers with the BLAST and SILVAngs outputs which contain metrics on sequence match.

Line 475. Why was rarefaction not implemented? Surely if you are comparing between sites and timepoints you would need the same number of reads across samples so as not to inflate diversity estimates. Please provide species accumulation curves by sequence depth for each site/timepoint (i.e. the combined triplicate since they are being merged) to assess whether accumulation has plateaued and therefore whether rarefaction is required or not.

As outlined in the reference we cited (McMurdie & Holmes 2014), rarefaction would be the correct strategy in this case. In addition, all samples had good read depth, and rarefaction curves showed samples to be plateauing. Consequently, we have utilised the whole dataset, focusing on read proportions, rather than focusing on just a fraction of the data generated and using subsampled read counts.

Reference:

McMurdie, P. J. & Holmes, S. Waste Not, Want Not: Why Rarefying Microbiome Data Is Inadmissible. *PLoS Comput. Biol.* 10, e1003531 (2014).

We have added these plots into the supplementary materials (supplementary figure 5).

We have also amended the text where we highlight our strategy, which now reads:

Line 523-525: “Rarefaction was not implemented due to its documented limitations⁶⁵, in addition to all samples showing adequate diversity saturation with appropriate read depths achieved (supplementary figure 5).”

Line 477. Can you explain how your normalised read counts are an appropriate proxy for abundance data? Would the read count per species not be affected by primer amplification biases (particularly with a two-step PCR) and sequencing depth? Many metabarcoding studies have reported that the observed proportion of species from eDNA metabarcoding data (in contrast to targeted species-specific assays) do not reflect the true proportion in the environment. For example, this recent study - Shelton et al. 2022; <https://esajournals.onlinelibrary.wiley.com/doi/full/10.1002/ecy.3906> found that “that simple tabulations of proportions of amplicon-sequence reads are likely to provide misleading inferences due amplification bias”. The above authors needed to calibrate the data to yield estimates of proportional contributions of each of the taxa that reflect the original biological sample prior to PCR.

Recently, Gold et al. 2023 (<https://journals.plos.org/plosone/article?id=10.1371/journal.pone.0285674>) reported that “metabarcoding datasets are strongly affected by (1) deterministic amplification biases during PCR and (2) stochastic sampling of amplicons during sequencing—both of which we can model—but also by (3) stochastic sampling of rare molecules prior to PCR, which remains a frontier for quantitative metabarcoding”.

The overarching messages emerging from the two papers from the Kelly lab are focused on absolute quantification and stochastic PCR sampling of rare molecules amongst technical replicates in the PCR process. Here, to reduce biases associated with stochastic PCR sampling, we use three ecological replicates per sampling site and complement with triplicated technical replication during the first amplification phase of the PCR reaction. In doing so, we endeavour to maximise ecological coverage and minimise poor subsampling of the rare biosphere (c.f. ASV rarefaction plots (supplementary figure 5)). Further, the Shelton et al. reference is referring to absolute quantification of species; a process that can only be achieved with very careful, species by species quantitative calibration, such as qPCR, or very intensive and elegant mock communities (Bista et al., 2018) that would be impossible to achieve at the ecosystem scale. Importantly, in the present manuscript, we have based all our analyses on the relative abundance of individual ASVs, over time and space, not the relative abundance of ASVs to each other. This means that the confounding effect of different mitochondrial copy numbers and different numbers of tandem repeats within ribosomal nuclear genes, PCR bias at the genome level etc. across different species does not confound our ecological interpretations. Interestingly, at least some of the analyses presented in Shelton et al. appears to be focused on quantification of genomic DNA, when in actual fact, any species-by-species quantification should be performed on the gene itself. Otherwise, any biases associated with gene copy numbers will not be captured in the modelling exercise.

Reference:

Bista, I., Carvalho, G. R., Tang, M., Walsh, K., Zhou, X., Hajibabaei, M., ... & Creer, S. (2018). Performance of amplicon and shotgun sequencing for accurate biomass estimation in invertebrate community samples. *Molecular ecology resources*, 18(5), 1020-1034.

Line 480. Please explain how you can use quantitative abundance data from eDNA metabarcoding for Shannon index values?

Added:

Line 545-548: “‘Vegan’ v 2.4.2⁶⁷ was used to calculate Shannon index values for the 18S and COI markers using the normalised read counts for the proportion an ASV contributes to a community and number of ASVs as a proxy for species number.”

Also why did you choose not to extend this to the 12S fish ASVs?

Shannon index was not used for the 12S because, as it had good species identification, we felt that it was more intuitive. That, and it made no difference to the trends if we used species index or Shannon index.

Line 504. Is the multivariate analysis (i.e. Bray-Curtis mMDS plots) based on normalised read count data for all three markers, or is the 12S data being treated different as it was for the alpha diversity analysis?

We have now added the following text to the 'statistical analysis' section:

Line 573-574: "Normalised read count were used as the input for the PERMANOVA and NMDS plots for all markers."

Again, I am concerned about the quantitative aspect whereby you are inputting uncalibrated metabarcoding data as a proxy of varying abundance between sites. Are you seeing the same site variation patterns (i.e. distance from the lake) if you convert your data to presence-absence and conduct Jaccard nMDS plots?

We thank the reviewer for their comment with regards quantitation of eDNA as this is a controversial subject within our field. However, the overall trends do remain robust whether we are using presence/absence or abundance in the analyses. Please see the reanalyses conducted and presented in the general comments section.

REVIEWERS' COMMENTS

Reviewer #2 (Remarks to the Author):

Perry et al. have thoroughly responded to all of the reviewer comments with subsequent revisions across the paper. I appreciate the significant time and effort they have put into additional analyses - particularly in regards to the abundance and rarefaction queries. The resulting paper is concise and informative and I am very happy to recommend it for publication. Congratulations to the authors and I look forward to seeing it in print.